# The local mechanostructural properties of protein cargoes regulate nucleocytoplasmic transport

**Rafael Tapia-Rojo** [1,2,4] ✉, **Natalie Milmoe**[1,2,4], **Patricia Paracuellos** [1,2], **Brian Lally**[1,2], **Cristina Escalona-López**[1,2], **Laura Masino** [3], **Jenny Gehlen** [1,2], **Jane Walker**[1,2] & **Sergi Garcia-Manyes** [1,2] ✉

The nuclear pore complex regulates nucleocytoplasmic transport. It was recently shown that the global mechanical stability of proteins regulates their nuclear import rate. On the basis of these findings, we hypothesize that the main principles governing protein translocation through narrow biological pores—in which locally unstructured and unfolded regions determine cargo orientation and translocation kinetics—can help rationalize protein trafficking across the nuclear pore complex. Inspired by single-molecule studies showing that proteins exhibit different mechanical stability when pulled from different termini, here we show that the rate of both nuclear import and export is enhanced when the translocating protein is threaded through the nuclear pore from the specific region exhibiting lower local nanomechanical stability and increased structural disorder. We demonstrate this for a range of model proteins with different folds and stabilities by combining single-molecule magnetic tweezers with single-cell optogenetic experiments, complemented by steered molecular dynamics simulations and biochemical binding assays. Our bioinformatics survey then shows that in human transcription factors, the termini containing the nuclear localization signal sequence exhibit a higher degree of structural disorder. We propose that protein orientation might offer an additional layer of structural and mechanical control of the kinetics of nuclear transport.

During translocation across biological membranes, or during proteolysis[1] and disaggregation[2], proteins undergo transient unfolding as they encounter narrow pores that are often too small to accommodate folded protein domains[3,4]. Despite the inherent structural and functional specificities of biological pores as diverse as those found in mitochondria[3], chloroplasts[5], toxins[6] or distinct proteasomal machineries[1], protein translocation across these highly specific molecular channels shares several general mechanistic traits in common[7,8]. In most cases, a dedicated ATPase machinery recognizes a pore-specific signalling tag and exerts mechanical force to aid in the unfolding process. The effectiveness of such translocation-coupled unfolding mechanisms is heavily dependent on the local structure and mechanical resistance of the protein region adjacent to the targeting sequence[8]. Once this local secondary structure motif unfolds, the rest of the polypeptide will rapidly denature, traverse and exit the pore and eventually refold—often assisted by evolved chaperone machineries.

[1]Single Molecule Mechanobiology Laboratory, The Francis Crick Institute, London, UK. [2]Department of Physics, Randall Centre for Cell and Molecular Biophysics, Centre for the Physical Science of Life and London Centre for Nanotechnology, King's College London, London, UK. [3]Structural Biology Science Technology Platform, The Francis Crick Institute, London, UK. [4]These authors contributed equally: Rafael Tapia-Rojo, Natalie Milmoe. ✉e-mail: rafael.rojo@kcl.ac.uk; sergi.garcia-manyes@kcl.ac.uk

Consequently, the position of the unstructured and mechanically vulnerable region along the protein backbone largely determines the direction of translocation[9,10].

Studding the nuclear envelope double membrane, the nuclear pore complex (NPC)—which acts as the main gateway in and out of the nucleus—is a highly specialized biological pore that shares few structural and mechanistic commonalities with other ATP-driven pores[11–13]. In particular, the pore diameter is very large (~40–70 nm)[12] and elastic[14–16], enabling the translocation of large protein cargoes[17] across a dense mesh of highly dynamic and intrinsically disordered nucleoporins (Nups) that create an effective permselective barrier[18–20]. Additionally, no obvious ATP-fuelled unfoldases have been identified to directly catalyse the nuclear shuttling process. Therefore, the mechanical unfolding of proteins would not be a priori necessary to enhance the protein's nuclear import rate. This is why it was somewhat surprising to find that the mechanical properties of proteins regulate their nuclear import rate across the NPC[21]. Additionally, for high-molecular-weight and structurally complex (poly)proteins, mechanically soft domains close to the nuclear localization signal (NLS) sequence markedly increase their nuclear translocation rate[22].

A unique characteristic of the mechanical unfolding of proteins is that the protein's mechanical stability depends on the direction from which the protein is pulled. This was revealed by single-molecule nanomechanical experiments in vitro[23,24], where the protein is generally held and stretched from both protein termini. When crossing a pore, by contrast, the unfolding pathway is substantially modified because the protein is only held by the terminus that first encounters the pore, leaving the other terminus unconstrained[8,25]. Therefore, the local stability of the protein's side that is first threaded through the pore is likely to play a fundamental role in determining the pore import kinetics. In fact, during protein translocation across mitochondria[3,26,27] or proteosomes[28], or even artificial nanopores[25], the presence of a loosely folded terminus with increased tendency to mechanically unravel determines the preferential site for pore translocation initiation.

This naturally led us to question whether, in the case of the NPC, the local structure and mechanical stability within a single protein domain also regulates its ability to translocate to the cell nucleus. We conjectured that the protein's local nanomechanical properties could determine their preferential orientation during nuclear translocation, and hypothesized that this could be a potential strategy to control the dynamics and directionality of nucleocytoplasmic transport of transcription factors (TFs), which continuously shuttle across the NPC.

We began by probing how the dynamics of nuclear entry are affected by the orientation (N-term to C-term or C-term to N-term) of a small translocating protein monomer by using an optogenetic tool that precisely controls nuclear/cytoplasmic localization on blue light illumination. We transiently transfected U2OS cells with a light-inducible nuclear export NLS-mCherry-*As*LOV2-NES (ref. 21) construct based on the LEXY probe[29] (Fig. 1a), which uses the light-induced conformational changes of the widely used LOV2 domain from *Avena sativa* to provide light-inducible spatiotemporal control of nucleocytoplasmic dynamics. In the absence of light, the nuclear export signal (NES) moiety is docked to the LOV2 domain, thereby unable to induce export. Consequently, the construct mostly accumulates in the nucleus. Upon blue light ($\lambda = 488$ nm) illumination, the LOV2 domain undergoes a fast, reversible conformational change that readily exposes the NES, resulting in the fast, quantitative migration of the optogenetic construct to the cytoplasm. Switching off the light again results in the redocking of the NES, and the concomitant exponential accumulation of the construct back into the nucleus, hallmarked by a characteristic time constant, $\tau$ ($k = 1/\tau$). To test the possible differential import kinetics when the optogenetic construct is imported into the nucleus from the N-term or the C-term, we used an optogenetic import-forward construct (where the mCherry fluorophore enters the NPC from its N-term) and developed an analogous import-reverse construct (where the C-term region of mCherry enters first) (Fig. 1b and Extended Data Fig. 1). Comparing the nuclear import kinetics of the NLS-mCherry-*As*LOV2-NES import-forward probe (Fig. 1c–e and Supplementary Fig. 1) with that of

**Fig. 1 | mCherry exhibits directional nuclear import due to the unravelling of a mechanically weak N-terminal intermediate. a**, Optogenetics assay to measure the time course of the nucleus-to-cell localization of protein constructs. Upon blue light illumination, the LOV2 domain exposes an NES sequence that allows the rapid export of the protein construct out of the nucleus. Switching off the blue light results in the NES docking back onto the LOV2 domain. The exposed NLS triggers the import of the optogenetic construct into the nucleus, enabling us to monitor its import kinetics and accumulation over time (recovery phase). Top: representative confocal images of U2OS cells at different time points. Scale bar, 10 μm. **b**, Schematics of the optogenetic constructs designed to investigate the differential nuclear import kinetics when the protein translocates from its N-terminus import forward or C-terminus import reverse. **c–e**, Nuclear import kinetics of mCherry when translocating from the N- or C-terminus. **c**, Representative confocal microscopy images of U2OS cells after 30 min in the recovery phase. **d**, Average time courses of the nucleus-to-cytoplasmic localization of the import-forward and import-reverse mCherry constructs. **e**, Nucleus-to-cytoplasm accumulation (top) and import rate (bottom) calculated from fits to the recovery time courses. Accumulation: N-term $K_e = 1.87 \pm 0.07$; C-term $K_e = 1.19 \pm 0.05$. Import rate: N-term $k_I = 3.22 \pm 0.15$ ks$^{-1}$; C-term $k_I = 2.06 \pm 0.09$ ks$^{-1}$. Significance levels for two-tailed Mann–Whitney non-parametric test. Accumulation, $P = 7.2 \times 10^{-12}$; import rate, $P = 3.5 \times 10^{-9}$. $n = 86$ (N-term); $n = 69$ (C-term) from $N > 3$ independent experiments. **f**, Schematics of the polyprotein construct used in our smMT experiments to measure the mechanical unfolding of a single mCherry protein. **g**, Representative unfolding trajectory of mCherry on a force ramp (1 pN s$^{-1}$). mCherry unfolds through a mechanically weak intermediate (blue arrow, marked as (1)) followed by a second mechanically stronger state (green arrow, marked as (2)). The corresponding increments in the contour length ($\Delta L_c$) are calculated using the freely jointed chain model for each individual unfolding step. **h**, Scatter plot showing the increase in contour length ($\Delta L_c$) versus unfolding force ($F_U$) for each unfolding event, together with the marginal probability distributions

for $\Delta L_c$ and $F_U$. The distribution of unfolding forces shows a set of mechanically weak events (blue) and stronger events (green). Data from $n = 50$ unfolding events measured on eight individual molecules. **i**, Representative trajectories of the unfolding pathway of mCherry obtained from SMD simulations, where the protein is held at a constant force of 350 pN applied to its N- and C-terminal amino acids. mCherry extends (~18 nm) quickly to an intermediate conformation (1) and then unravels its remaining structure (2). **j**, Representative snapshots of the force-induced conformations (1) and (2) visited by mCherry, showing that the weak intermediate state corresponds to the unfolding of its N-term region. **k**, Diagram of the mCherry protein (top) with the N-terminus in blue and the C-terminus in green. Scissors mark the position at which the new termini are generated to synthesize a circularly permutated protein (bottom). **l–n**, Nuclear import kinetics of mCherry when translocating from the N-terminus for both WT mCherry (data displayed above) and circ-p mCherry. **l**, Representative confocal microscopy images of U2OS cells after 30 min in the recovery phase. Scale bar, 10 μm. **m**, Average time courses of the nucleus-to-cytoplasmic localization of WT mCherry and circ-p mCherry being imported from the N-term (forward). **n**, Relative nuclear accumulation (top) and import rate (bottom) calculated from fits to the recovery time courses. Bars indicate mean. Accumulation, $K_e = 0.89 \pm 0.03$; import rate, $k_I = 1.30 \pm 0.05$ ks$^{-1}$. Significance levels for two-tailed Mann–Whitney non-parametric test. Accumulation, $P = 2.2 \times 10^{-16}$; import rate, $P = 4.4 \times 10^{-16}$. $n = 56$ (circ-p) from $N > 3$ independent experiments. **o**, Representative magnetic tweezers unfolding trajectory of circ-p mCherry on a force ramp (1 pN s$^{-1}$). Circ-p mCherry unfolds through an intermediate that is mechanically stronger than that of WT mCherry (blue arrow, marked as (1)). **p**, Scatter plot showing the increase in contour length ($\Delta L_c$) versus unfolding force ($F_U$) for the mechanically weakest circ-p mCherry unfolding event, and the associated marginal probability distributions for $\Delta L_c$ and $F_U$. Unfolding events for WT mCherry are shown as shaded circles for comparison. Data from $n = 59$ events on six molecules.

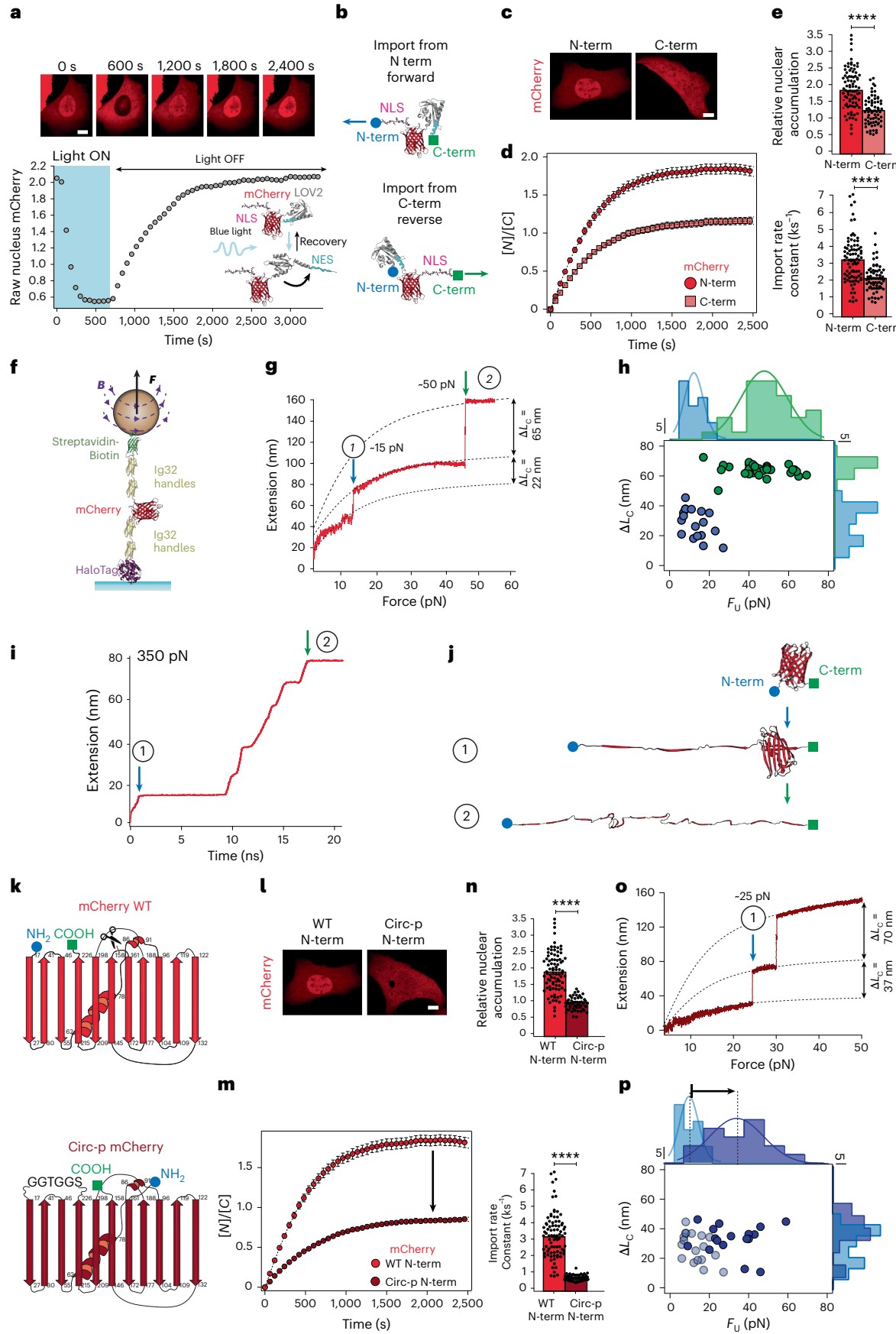

the mirroring construct, *As*LOV2-NES-mCherry-NLS (import-reverse) shows markedly different nuclear import kinetics (Fig. 1d,e), with the import-forward probe showing a significantly higher nuclear accumulation at equilibrium ($K_e = 1.87 \pm 0.07$) and import rate ($k_I = 3.22 \pm 0.15$ ks$^{-1}$) than the import-reverse probe ($K_e = 1.19 \pm 0.05$, $k_I = 2.06 \pm 0.09$ ks$^{-1}$; Fig. 1e). The same asymmetric nuclear import was also observed in HeLa and NIH 3T3 cells (Supplementary Fig. 2). These results demonstrate a substantial degree of asymmetry in the nuclear import rate according to the translocation orientation of the optogenetic probe.

To rationalize the molecular origin of such asymmetric behaviour, we used single-molecule magnetic tweezers (smMT)[30] to probe the mechanical stability of the mCherry protein monomer (Fig. 1f). Applying a force-ramp protocol in which the force applied to the protein is linearly increased over time (1 pN s$^{-1}$)[31] enabled us to sample the unfolding pathway, which showed a hierarchical mechanical pattern involving a first mechanically weak event characterized by a step-wise increase in length of ~18 nm ($F = 15 \pm 3$ pN), followed by a mechanically stronger event (~60 nm; $F = 50 \pm 5$ pN; Fig. 1g,h). The associated contour length values for both unfolding steps observed in the force-ramp experiments were calculated with the freely jointed chain model of polymer elasticity, resulting in $\Delta L_c = 22$ nm and $\Delta L_c = 65$ nm, respectively (Fig. 1g,h).

These results demonstrate that mCherry unfolds first via an intermediate of low mechanical stability. However, due to the intrinsic nature of smMT experiments (where end-to-end protein extension changes are monitored), one cannot unambiguously assess the specific region across the mCherry structure that unravels first, underpinning the low-stability mechanical intermediate. To provide a sub-molecular picture of the unfolding pathway that enables us to map out the position of the main rupture events within the protein structure identified in the single-molecule nanomechanical experiments, we conducted steered molecular dynamics (SMD) simulations[32,33]. Pulling the mCherry monomer from both termini (that is, mimicking the magnetic tweezers experiments) under constant-force conditions (350 pN) resulted in the first quick unfolding event (Fig. 1l(1)) reminiscent of a mechanically labile intermediate conformation, followed by a series of consecutive rupture events occurring at a much later time, which ultimately led to a completely unfolded and extended protein (Fig. 1l(2)). This sequence of rupture events matches the unfolding pathway observed in the single-molecule unfolding trajectories (Supplementary Fig. 3). Specifically, the first unfolding event, characterized by a protein extension of ~18 nm (corresponding to $\Delta L_c = 22$ nm) (1), relates to the complete peeling of the first three N-terminal β-sheets (residues 1–56) from the main protein core before the protein undergoes complete unravelling at a much later time (2) through the sequential disruption of its remaining structure (Fig. 1j). The combination of smMT experiments and SMD simulations rationalize the optogenetic experiments in single cells, suggesting that in the import-forward construct, mCherry unfolds its N-terminus mechanical intermediate first, thereby generating an unstructured region that increases the overall nuclear import rate compared with the import-reverse construct, where the protein enters the pore from its more rigid C-terminus. As a result, the structural and mechanical anisotropies imposed by the N-terminus unfolding intermediate in mCherry seems to determine its enhanced nuclear import kinetics when entering the NPC from this terminus.

In an effort to probe whether perturbing the structural (and mechanical) architecture of the unfolding termini affects mCherry's import rate, we engineered a circular permutant of the mCherry protein (circ-p mCherry), where the original termini are linked together and the new termini are created in a different position within the protein structure[34], namely, in positions 193 and 194 (Fig. 1k). Such a structural rearrangement results in a change in the protein's unfolding pathway, which we conjectured would have an effect on the measured nuclear import rate of circ-p mCherry with respect to that of the wild-type (WT) mCherry in the optogenetic assay. Our results show that although the

import hierarchy is maintained (the protein translocates faster from the N-terminus than from the C-terminus) (Supplementary Fig. 4), the import rate of circ-p mCherry from its N-terminus is significantly lower than that of the WT mCherry form (Fig. 1l–n), suggesting that the structure of the new circ-p mCherry terminus poses more resistance to unfolding. Probing the stability of the circ-p mCherry using smMT with the same force-ramp strategy demonstrated that the first unfolding event undergoes a marked shift in force, increasing from ~15 pN to ~25 pN (Fig. 1o,p). This confirmed our hypothesis that altering the local mechanical stability of the protein region that enters the NPC first affects its import rate into the nucleus.

We next queried whether the presence of a clear and long-lived low-mechanical-stability unfolding intermediate is required to induce protein directionality during nuclear import, or if proteins exhibiting a global all-or-none, two-state unfolding pathway can still show nuclear import anisotropy as a consequence of subtle differences in the local mechanical stability (a proxy for local structure) between their N- and C-termini. To test this hypothesis, we used the extensively characterized 27th Ig domain of titin[35], Ig27, as a model system (Fig. 2a). As expected, when pulled in our magnetic tweezers (Fig. 2b) Ig27 unfolds at high forces (~100 pN) in a single ~27-nm step (pulling rate, 1 pN s$^{-1}$), bringing the protein close to its contour length (Fig. 2c,d)[36]. This two-state unfolding mechanism can be easily ascribed to the sudden, cooperative rupture of the six hydrogen bonds (the 'mechanical clamp') between β-strands A–A′ and G, after which the unfolding process becomes downhill, in agreement with earlier work[32] (Fig. 2e). To study the impact of Ig27 orientation on the dynamics of nuclear import across the NPC in U2OS cells, we used the same optogenetic approach (Fig. 2f) and compared the dynamics of nuclear entry for both import forward, NLS-X-mCherry-*As*LOV2-NES, and import reverse, *As*LOV2-NES-mCherry-X-NLS, optogenetic constructs, generally modified to contain a specific mechanical reporter protein of interest (X); in this case, X = Ig27 (Extended Data Fig. 2). We found that the nuclear accumulation and the rate of nuclear import for the NLS-Ig27-mCherry-*As*LOV2-NES import-forward probe (where the Ig27 is imported from its N-terminus) was significantly higher ($K_e = 1.34 \pm 0.05$, $k_I = 1.72 \pm 0.08$ ks$^{-1}$) than that of the import-reverse *As*LOV2-NES-mCherry-Ig27-NLS construct ($K_e = 1.03 \pm 0.04$, $k_I = 0.99 \pm 0.05$ ks$^{-1}$), whereby the Ig27 monomer enters the NPC from its C-terminus instead (Fig. 2g–i and Supplementary Fig. 5). Treating the cells with 20 nM of leptomycin B (which blocks nuclear export by inhibiting CMR1 (ref. 37)) confirmed that Ig27 imported from the N-terminus significantly faster than from the C-terminus (Supplementary Fig. 6). The same asymmetric nuclear import was also observed in HeLa and NIH 3T3 cells (Supplementary Fig. 7).

To examine the sub-molecular origin of these orientation-dependent nuclear import dynamics and, most importantly, to mimic the geometry and unfolding pathway followed by proteins while importing across pores, we conducted a new set of SMD simulations to explore the local mechanical stability of the N- and C-termini independently. These SMD simulations were specifically designed to pull the protein from only one terminus (N-term pulling or C-term pulling), which is a stretching geometry that cannot be accessed through single-molecule force spectroscopy experiments. In these simulations, we constricted the majority of the protein (Fig. 2j)—leaving only the N- or C-terminus β-strand freely unconstrained—and we mechanically pulled from the constrained protein structure and the terminal amino acid in the unconstrained N- or C-terminal strand simultaneously. This pulling strategy forces the selective unfolding of each independent terminal, thereby mimicking the pulling geometry it might experience on directional translocation through the NPC. By averaging several ($n = 10$; Fig. 2k and Supplementary Fig. 8) individual SMD trajectories, we compared the force required to unfold each Ig27 terminal β-strand, and found that the unfolding force required to unravel the C-terminus (involving β-strand G) is 67% higher than that needed to unfold the protein

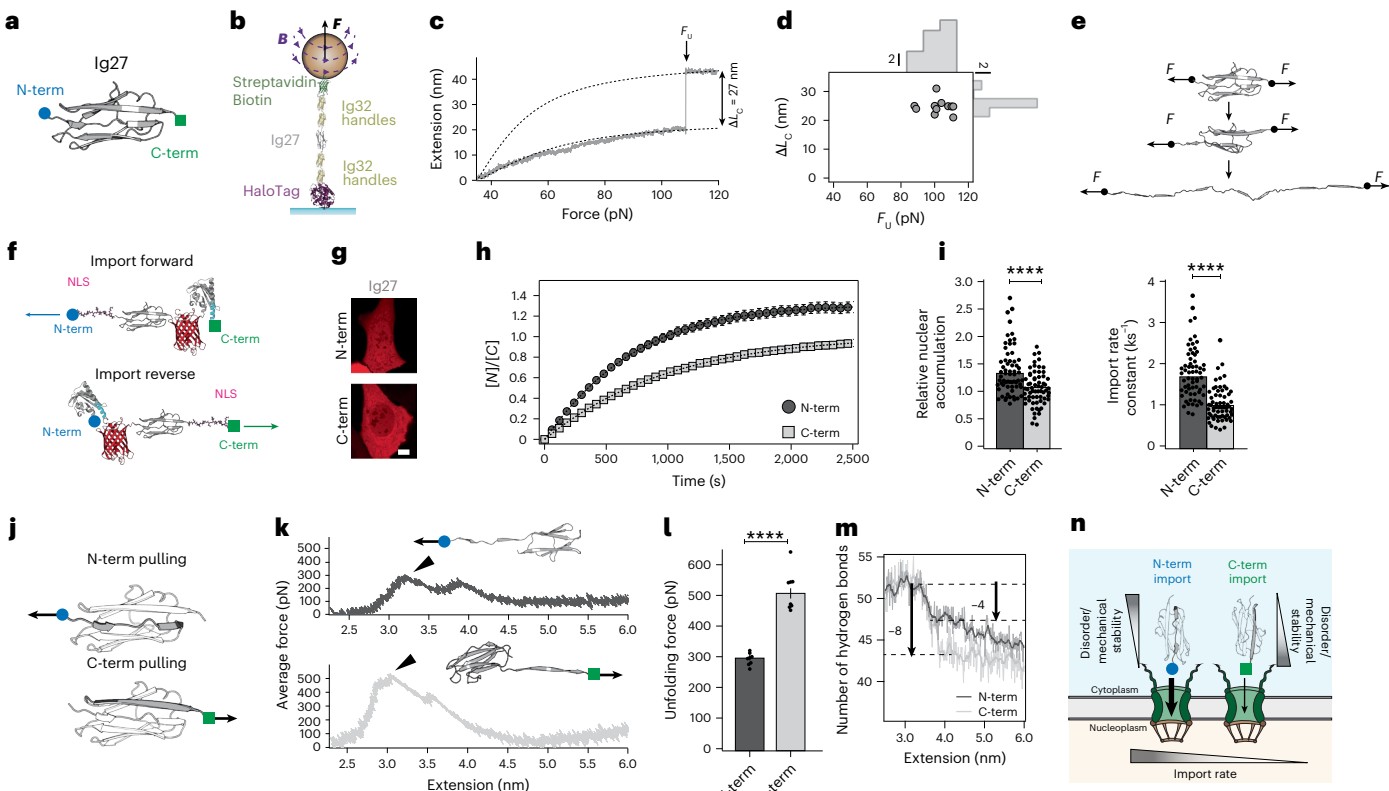

**Fig. 2 | Mechanical and structural anisotropies of both termini of Ig27 leads to its asymmetric nuclear import. a**, Structure of the titin Ig27 domain, highlighting its N-(blue) and C-(green) termini. **b**, Schematics of the smMT nanomechanics experiment. **c**, Representative magnetic tweezers trajectory in a force ramp (1 pN s⁻¹), showing that Ig27 unfolds cooperatively in a single event at a force of −110 pN. **d**, Scatter plot of the increase in contour length ($\Delta L_c$) versus unfolding force ($F_U$) for each unfolding event, together with the marginal probability distributions for $\Delta L_c$ and $F_U$. Data from $n = 11$ unfolding events measured on 11 molecules. **e**, Ig27 unfolds cooperatively on rupture of its mechanical clamp. **f**, Schematics of the import-forward and import-reverse constructs designed to compare the nuclear import kinetics when the Ig27 is imported from its N- or C-terminus. **g–i**, Differential nuclear import kinetics of Ig27 when translocating from its N- or C-terminus. **g**, Representative confocal images of U2OS cells after 30 min in the recovery phase. Scale bar, 10 μm. **h**, Average time courses of the nucleus-to-cytoplasm localization of the import-forward and import-reverse optogenetic constructs containing X = Ig27. **i**, Relative nuclear accumulation (left) and import rate (right) calculated from fits to the recovery time courses. Accumulation: N-term $K_e = 1.34 \pm 0.05$;

C-term $K_e = 1.03 \pm 0.04$. Import rate: N-term $k_I = 1.72 \pm 0.08$ ks⁻¹; C-term $k_I = 0.99 \pm 0.05$ ks⁻¹. Significance levels for two-tailed Mann–Whitney non-parametric test. Accumulation, $P = 2.3 \times 10^{-5}$; import rate, $P = 2.3 \times 10^{-12}$. $n = 64$ (N-term); $n = 64$ (C-term) from $N > 3$ independent experiments. **j**, Schematics of the pulling protocol to independently probe the mechanical stability of the N-term and C-term regions of I27 using SMD simulations. **k**, Average force-extension curves for the N-term (top, blue) and C-term (bottom, green) pulling modes, showing the representative snapshot in which the N-term (A–A′ strand) or C-term (G strand) has been respectively unfolded. **l**, Distribution of unfolding forces corresponding to the unfolding of Ig27 from its N-term (blue) or C-term (green) obtained from ten SMD simulations (pulling speed, $v = 1$ nm ns⁻¹). $P = 4.8 \times 10^{-5}$. **m**, Evolution of the number of hydrogen bonds in Ig27 under the N-term (blue) or C-term (green) pulling modes. The higher mechanical stability of the C-term Ig27 region can be attributed to the higher number of hydrogen bonds that are disrupted on unravelling (eight in the C-term versus four in the N-term). **n**, Schematics of our proposed mechanism for orientation-dependent protein import, where the mechanically weaker protein terminus exhibits a higher nuclear import rate.

from its N-terminus (β-strand A–A′) (Fig. 2l). Specifically, the higher mechanical stability of the Ig27's C-terminus is directly correlated to the higher number of native hydrogen bonds that need to be broken to unravel this N-terminal β-strand (eight hydrogen bonds) compared with those disrupted in the N-terminal (four hydrogen bonds) (Fig. 2m and Supplementary Fig. 9). In summary, the lower local mechanical stability of the protein's N-terminus is likely to underpin the higher nuclear import rate measured in the optogenetic experiments when Ig27 is imported through the NPC from its N-terminus (Fig. 2n). This was further supported by adding an unstructured glycine-serine (GS)₂₅ tag[22] to either the N- or C-terminus of Ig27, which resulted in the abolished asymmetry of nuclear import (Extended Data Fig. 3). This suggested that the presence of an unstructured and mechanically loose peptide next to the specific terminus containing the NLS masks the local mechanical stability of Ig27, thereby removing import directionality.

These observations raised the question of whether the differences in nuclear import that we observed when Ig27 was imported through its

N- and C-termini could be explained in terms of a different NLS accessibility that alters its binding affinity with importin α3 (which is the isoform that binds the cMyc NLS[38] (Fig. 3a), and is expressed in U2OS, HeLa and NIH 3T3 fibroblasts; Supplementary Fig. 10). To explore this, we conducted isothermal titration calorimetry (ITC), which showed a near-identical binding isotherm for NLS-Ig27 and Ig27-NLS to importin α3, with associated dissociation constants of $K_D = 626 \pm 60$ nM and $K_D = 605 \pm 55$ nM, respectively (Fig. 3b,c, Supplementary Fig. 11 and Supplementary Table 1). Similarly, pull-downs of His-tagged importin α3 and strep-tagged N- or C-term NLS-Ig27 showed similar binding levels, regardless of the orientation of the NLS (Extended Data Fig. 4). These experiments concluded that the orientation-dependent import rate observed in our optogenetic experiments cannot be explained in terms of a preferential binding of the NLS sequence placed on a specific Ig27 terminus to importin α3.

We then speculated that the difference in the import rates that we observed when proteins were threaded into the pore from different

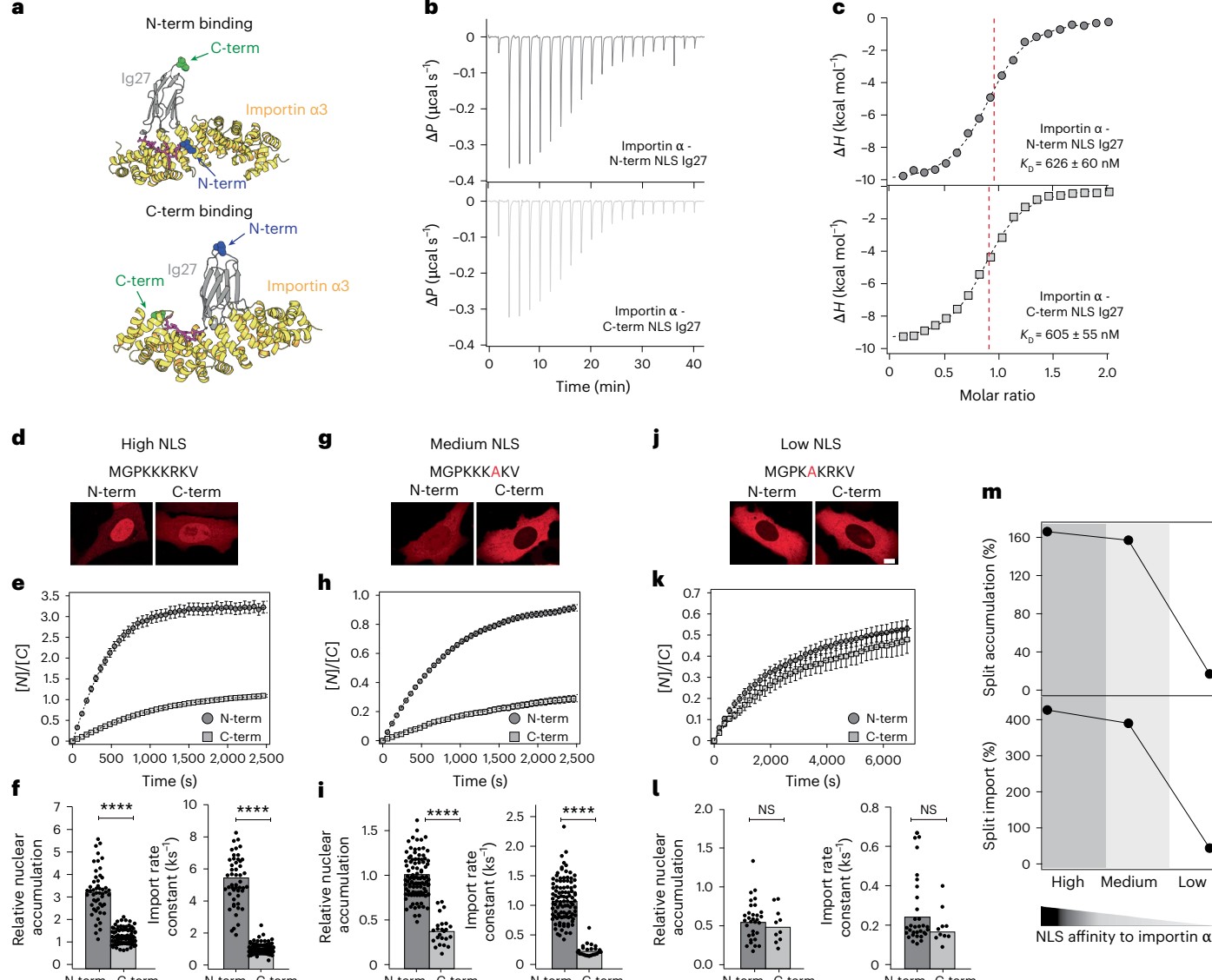

**Fig. 3 | NLS orientation does not affect cargo binding to importin α3, whereas increased NLS affinity to importin α3 enhances asymmetric nuclear import. a**, AlphaFold3 structural model of cMyc-NLS- Ig27 (top) and Ig27-cMyc-NLS (bottom) bound to importin α3. **b**,**c**, Titration of cMyc-NLS- Ig27 (top) and Ig27-cMyc-NLS (bottom) (17 µM) into importin α3 (170 µM). **b**, Thermogram for injections of cMyc-NLS-Ig27 (top) and Ig27-cMyc-NLS (bottom). Δ*P*, differential power. **c**, Incremental enthalpy changes, corrected for heat of dilution. Data were fitted using a one-site binding model. **d**–**f**, Nuclear import kinetics of Ig27 when translocating from its N- or C-terminus in the optogenetic constructs containing the SV40 NLS sequence that exhibits high affinity to importin α. **d**, Representative confocal images of U2OS cells after 30 min, in the recovery phase. Scale bar, 10 µm. **e**, Average time courses of the nucleus-to-cytoplasm localization of the import-forward and import-reverse optogenetic constructs (X = Ig27) modified to harbour the high-affinity NLS. **f**, Relative nuclear accumulation (left) and import rate calculated from fits to the recovery time courses. Bars indicate mean. Accumulation: N-term $K_e = 3.22 \pm 0.14$; C-term $K_e = 1.21 \pm 0.04$. Import rate: N-term $k_1 = 5.38 \pm 0.04$ ks$^{-1}$; C-term $k_1 = 1.02 \pm 0.04$ ks$^{-1}$. Significance levels for two-tailed Mann–Whitney non-parametric test. Accumulation, $P = 8.9 \times 10^{-16}$; import rate, $P = 4.4 \times 10^{-16}$. $n = 51$ (N-term); $n = 88$ (C-term) from $N > 3$ independent experiments. **g**–**i**, Nuclear import kinetics of Ig27 when translocating from its N- or C-terminus in the optogenetic constructs containing the SV40 NLS sequence that exhibits medium affinity to importin α. **g**, Representative confocal images of U2OS cells after 30 min, in the recovery phase. Scale bar, 10 µm. **h**, Average time courses of

the nucleus-to-cytoplasm localization of the import-forward and import-reverse optogenetic constructs (X = Ig27) modified to harbour the medium-affinity NLS. **i**, Relative nuclear accumulation (left) and import rate calculated from fits to the recovery time courses. Bars indicate mean. Accumulation: N-term $K_e = 0.98 \pm 0.05$; C-term $K_e = 0.38 \pm 0.02$. Import rate: N-term $k_1 = 1.08 \pm 0.03$ ks$^{-1}$; C-term $k_1 = 0.23 \pm 0.02$ ks$^{-1}$. Significance levels for two-tailed Mann–Whitney non-parametric test. Accumulation, $P = 1.5 \times 10^{-13}$; import rate, $P = 4.4 \times 10^{-14}$. $n = 100$ (N-term); $n = 24$ (C-term) from $N > 3$ independent experiments. **j**–**l**, Nuclear import kinetics of Ig27 when translocating from its N- or C-terminus in the optogenetic constructs containing the SV40 NLS sequence that exhibits low affinity to importin α. **j**, Representative confocal images of U2OS cells after 70 min, in the recovery phase. Scale bar, 10 µm. **k**, Average time courses of the nucleus-to-cytoplasm localization of the import-forward and import-reverse optogenetic constructs (X = Ig27) modified to harbour the low-affinity NLS. **l**, Relative nuclear accumulation (left) and import rate calculated from fits to the recovery time courses. Bars indicate mean. Accumulation: N-term $K_e = 0.57 \pm 0.04$; C-term $K_e = 0.51 \pm 0.07$. Import rate: N-term $k_1 = 0.26 \pm 0.03$ ks$^{-1}$; C-term $k_1 = 0.19 \pm 0.03$ ks$^{-1}$. $n = 33$ (N-term); $n = 10$ (C-term) from $N > 3$ independent experiments. Significance levels for two-tailed Mann–Whitney non-parametric test. Accumulation, $P = 0.62$; import rate, $P = 0.19$. $n = 31$ (N-term); $n = 10$ (C-term) from $N > 3$ independent experiments. **m**, Relative N-to-C-term split for accumulation (top) and import rate (bottom) for the high-, medium- and low-affinity NLS sequences, calculated as the percentage difference in the ratio between the N-term and C-term accumulation/import rate.

directions could be linked to the strength of the binding affinity of the NLS sequence to importin or the specific NLS sequence. To test this hypothesis, we modified our import-forward and import-reverse optogenetic constructs and replaced the constitutive cMyc-NLS with a 'high-affinity', a 'medium-affinity' and a 'low-affinity' version of the SV40 NLS[15] (Fig. 3d,g,j). For each NLS sequence, we compared the rate of nuclear translocation of Ig27 when imported from its N- and C-termini. In all cases, we observed the same trend as when using the cMyc-NLS of the original import-forward optogenetic probe, namely, that the N-terminus exhibits a faster import rate than the C-terminus (Fig. 3d–l and Supplementary Fig. 12). Interestingly, the split between the import rates between the N- and C-terminus increases with the strength of the NLS (Fig. 3m). These results suggest that the protein orientation effect that we uncovered is independent of the NLS sequence, but that this effect is enhanced with the strength of the NLS and transport is made increasingly active.

We next tested the generality of the cargo-orientation dependency of the rate of nuclear entry that we uncovered for the mCherry β-barrel protein and the Ig27 domain (also rich in β-sheet content) and expanded our assay to other model protein cargoes with different folds, structures and mechanical stabilities. We started with protein L, exhibiting an α/β fold and intermediate mechanical stability (Extended Data Fig. 5)[39] (~50 pN), as revealed by smMT experiments (Extended Data Fig. 5a–d). SMD simulations demonstrate that when specifically pulled from the C-term, protein L exhibits 22% higher mechanical stability than when pulled from the N-term (Extended Data Fig. 5e–g and Supplementary Fig. 13). These results are consistent with cellular optogenetic experiments, showing that protein L shows an enhanced nuclear translocating rate when imported from its mechanically weaker N-term (Extended Data Fig. 5h–j and Supplementary Fig. 14). We then probed the nuclear import orientation dependency of the spectrin R16 domain (Extended Data Fig. 6)[40], a mechanically soft protein[40] (unfolding at ~18 pN in our smMT nanomechanical experiments; Extended Data Fig. 6a–d) rich in α-helix content. In contrast with mCherry, Ig27 and protein L, the N-term region of R16 is mechanically more stable (36%) than its C-term, as revealed by SMD simulations (Extended Data Fig. 6e–g and Supplementary Fig. 15), again rationalizing our optogenetic experiments that show a much higher rate when R16 is imported from its C-terminus (Extended Data Fig. 6h–j and Supplementary Fig. 14). In summary, for each tested protein, the results from the single-cell optogenetic experiments correlate well with SMD simulations, exploring the local mechanical stability of the individual termini of the protein. In each protein, the nuclear import rate is inversely proportional to the local mechanical stability of each specific protein terminus.

These findings provide a means to potentially predict and control the kinetics of protein translocation across the NPC through the regulation and design of the nanomechanical stability of the localized secondary structure motifs present in specific positions within the protein structure, namely, those close to the protein's termini. To begin to explore this possibility, we turned to the two-domain Spy0128 protein from *Streptococcus pyogenes*. Spy0128 contains an intramolecular covalent isopeptide bond in each domain (Lys[36]–Asn[168] and Lys[179]–Asn[303] in its N and C domains, respectively) that physically connect, in both domains[41], their respective termini (Fig. 4a). This locks the protein in its natively folded state and prevents its extension. Consequently, as demonstrated with single-molecule atomic force microscopy experiments[42], Spy0128 is mechanically ultrastable. Such a unique protein architecture, where both termini are essentially glued together, should, in principle, prevent the creation of partially unfolded regions close to the termini, implying that Spy0128 should exhibit a symmetric behaviour when imported to the nucleus through the NPC from its N- or C-terminus. To test this hypothesis, we compared the import rate of the Spy0128 protein when inserted in the import-forward and import-reverse optogenetic probes. As expected, the import rate of Spy0128 when threaded from the N-terminus is slow and indistinguishable from that when imported from the C-terminus (Fig. 4b–d and Supplementary Fig. 16), in line with the hypothesis that no partial unfolding is possible and, consequently, that the protein traverses the pore folded irrespective of its orientation. Formation of the Spy0128 isopeptide bonds is catalysed by nearby glutamic acid residues, Glu117 and Glu258, in the N- and C-domains, respectively. Specific point mutations of these residues, namely, E117A and E258A, abrogate isopeptide bond formation, resulting in protein mutants that become suddenly extensible on mechanical unfolding[21,42]. We postulated that selectively preventing the formation of each isopeptide bond would result in faster and asymmetrical translocation of the related Spy0128 mutant when imported through the NPC from the newly extensible terminus.

We compared the nuclear translocation kinetics of the import-forward and import-reverse optogenetic probes containing the Spy0128 E117A mutant. We found that in contrast to Spy0128$_{WT}$, the E117A N-terminal mutant exhibits a significantly higher nuclear accumulation (N-term $K_e = 0.66 \pm 0.02$, compared with the C-term $K_e = 0.55 \pm 0.03$) and faster nuclear import rate ($k_I = 0.65 \pm 0.03$ ks$^{-1}$ compared with the C-term $k_I = 0.40 \pm 0.04$ ks$^{-1}$) when imported through the N-terminus (Fig. 4e–h and Supplementary Fig. 16). The Spy0128 E258A C-terminal mutant displayed the opposite effect, showing a higher nuclear accumulation ($K_e = 1.16 \pm 0.07$, compared with $K_e = 0.69 \pm 0.03$ for the N-term) and faster import rate ($k_I = 0.92 \pm 0.07$ ks$^{-1}$

**Fig. 4 | Asymmetric nuclear transport of Spy0128 and its variants. a**, Structure of Spy0128$_{WT}$, where the isopeptide bonds that render the protein inextensible are highlighted in cyan. **b–d**, Nuclear import kinetics of the Spy0128$_{WT}$ when imported from its N- or C-terminus. **b**, Representative confocal microscopy images of U2OS cells after 30 min of the recovery phase. Scale bar, 10 μm. **c**, Average time courses of the nucleus-to-cytoplasm localization of the Spy0128$_{WT}$ imported from either its N- or C-terminus. **d**, Relative nuclear accumulation (left) and import rate calculated from fits to the recovery time courses. Bars indicate mean. Accumulation: N-term $K_e = 0.54 \pm 0.02$; C-term $K_e = 0.58 \pm 0.03$. Import rate: N-term $k_I = 0.38 \pm 0.01$ ks$^{-1}$; C-term $k_I = 0.35 \pm 0.02$ ks$^{-1}$. Significance levels for two-tailed Mann–Whitney non-parametric test. Accumulation, $P = 0.38$; import rate, $P = 0.69$. $n = 81$ (N-term); $n = 51$ (C-term) from $N > 3$ independent experiments. **e**, Structure of Spy0128 E117A. The abrogation of the N-term isopeptide bond allows N-term region unfolding, whereas the C-term region remains inextensible. **f–h**, Nuclear import kinetics of the Spy0128 E117A when imported from its N- or C-terminus. **f**, Representative confocal images of U2OS cells after 30 min of the recovery phase. Scale bar, 10 μm. **g**, Average time courses of the nucleus-to-cytoplasm localization of the import-forward and import-reverse optogenetic constructs harbouring the Spy0128 E117A protein cargo.

**h**, Relative nuclear accumulation (left) and import rate (right) calculated from fits to the recovery time courses. Bars indicate mean. Accumulation: N-term $K_e = 0.66 \pm 0.02$; C-term $K_e = 0.55 \pm 0.03$. Import rate: N-term $k_I = 0.65 \pm 0.03$ ks$^{-1}$; C-term $k_I = 0.40 \pm 0.04$ ks$^{-1}$. Significance levels for two-tailed Mann–Whitney non-parametric test. Accumulation, $P = 2.2 \times 10^{-2}$; import rate, $P = 5.0 \times 10^{-8}$. $n = 75$ (N-term); $n = 27$ (C-term) from $N > 3$ independent experiments. **i**, Structure of Spy0128 E258A. The abrogation of the C-term isopeptide bond enables the unfolding of the C-term region, whereas the N-term region remains folded. **j–l**, Nuclear import kinetics of the Spy0128 E258A when imported from its N- or C-terminus. **j**, Representative confocal images of U2OS cells after 30 min of the recovery phase. Scale bar, 10 μm. **k**, Average time courses of the nucleus-to-cytoplasm localization of the import-forward and import-reverse constructs containing Spy0128 E258A. **l**, Relative nuclear accumulation (left) and import rate (right) calculated from fits to the recovery time courses. Accumulation: N-term $K_e = 0.69 \pm 0.03$; C-term $K_e = 1.16 \pm 0.07$. Import rate: N-term $k_I = 0.65 \pm 0.03$ ks$^{-1}$; C-term $k_I = 0.92 \pm 0.07$ ks$^{-1}$. Significance levels for two-tailed Mann–Whitney non-parametric test. Accumulation, $P = 5.1 \times 10^{-9}$; import rate, $P = 5.0 \times 10^{-3}$. $n = 50$ (N-term); $n = 35$ (C-term) from $N > 3$ independent experiments.

compared with the N-term $k_1 = 0.65 \pm 0.03\ \text{ks}^{-1}$) when imported from the C-terminus, where the absence of the isopeptide bond makes the protein extensible and, hence, able to partially unfold (Fig. 4i–l and Supplementary Fig. 16). In summary, our optogenetic experiments, where the nuclear localization tag (NLS) is placed adjacent to one of the protein termini, directly demonstrate that the mechanical stability of the local structure adjacent to the signal tag determines the rate of nuclear passage of the translocating protein, and that this can be selectively engineered to favour proteins to translocate across the NPC along a defined (N–C or C–N) backbone direction.

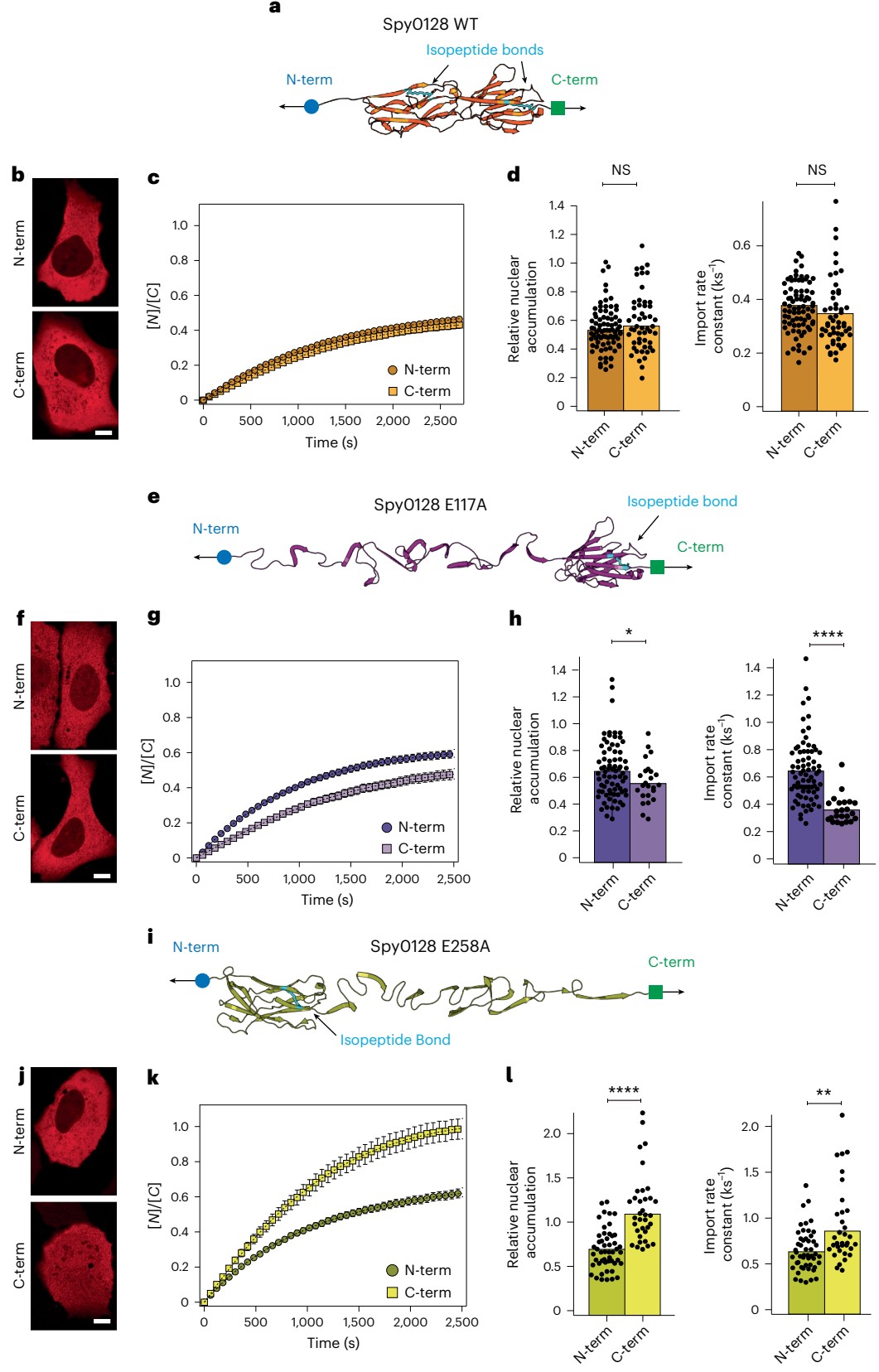

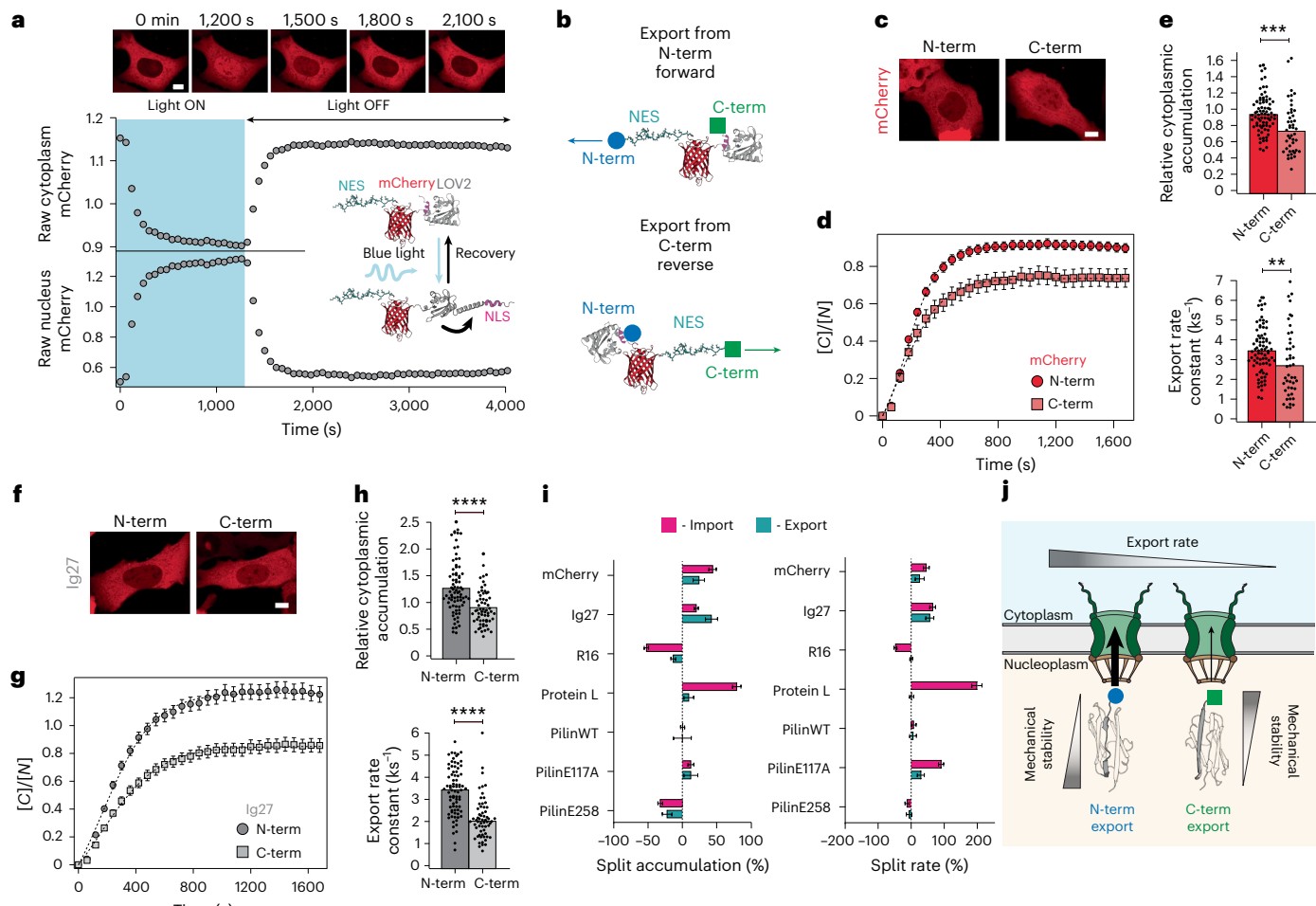

**Fig. 5 | Nuclear export kinetics are sensitive to the mechanical asymmetry of the translocating protein, but to a lower extent than nuclear import.** **a**, Optogenetics assay to measure the time course of the nucleus-to-cell localization of protein constructs. Upon blue light illumination, the export construct exposes an NLS sequence that allows the rapid import of the protein construct into the nucleus. Upon switching off the blue light (recovery phase), the NLS docks back onto the LOV2 domain, and the permanently exposed NES triggers the export of the protein out the nucleus, enabling monitoring its export kinetics and cytoplasmic accumulation. Top: representative confocal images of U2OS at different frames. Scale bar, 10 μm. **b**, Schematics of the constructs designed to investigate the differential nuclear export kinetics when the protein is exported from its N-terminus export forward or its C-terminus export reverse. **c–e**, Nuclear export kinetics of mCherry when translocating from its N- or C-terminus. **c**, Representative confocal images of U2OS cells after 25 min into the recovery phase. Scale bar, 10 μm. **d**, Average time courses of the nuclear-to-cytoplasmic localization of the export-forward and export-reverse optogenetic constructs containing mCherry. **e**, Relative cytoplasmic accumulation (top) and export rate (bottom) calculated from fits to the recovery time courses. Bars indicate mean. Accumulation: N-term $K_e = 0.95 \pm 0.03$; C-term $K_e = 0.76 \pm 0.05$. Export rate: N-term $k_E = 3.51 \pm 0.14$ ks$^{-1}$; C-term $k_E = 2.77 \pm 0.26$ ks$^{-1}$. Significance levels for two-tailed Mann–Whitney non-parametric test. Accumulation, $P = 1.1 \times 10^{-4}$;

export rate, $P = 3.3 \times 10^{-3}$. $n = 80$ (N-term); $n = 45$ (C-term) from $N > 3$ independent experiments. **f–h**, Nuclear export kinetics of Ig27 when exporting to the cytoplasm from its N- or C-terminus. **f**, Representative confocal images of U2OS cells after 25 min of the recovery phase. Scale bar, 10 μm. **g**, Average time courses of the nuclear-to-cytoplasmic localization of the export-forward and export-reverse optogenetic constructs containing the X = Ig27 cargo. **h**, Relative cytoplasmic accumulation (top) and export rate (bottom) calculated from fits to the recovery time courses. Bars indicate mean. Accumulation: N-term $K_e = 1.29 \pm 0.06$; C-term $K_e = 0.89 \pm 0.04$. Export rate: N-term $k_E = 3.34 \pm 0.13$ ks$^{-1}$; C-term $k_E = 2.11 \pm 0.12$ ks$^{-1}$. Significance levels for two-tailed Mann–Whitney non-parametric test. Accumulation, $P = 5.1 \times 10^{-7}$; export rate, $P = 6.2 \times 10^{-11}$. $n = 79$ (N-term); $n = 62$ (C-term) from $N > 3$ independent experiments. **i**, Split in accumulation (left; calculated as the percentage difference in the ratio between the nuclear (import) and cytoplasmic (export) accumulation between N- and C-terminal constructs) and import/export rate (right; calculated as the percentage difference ratio between N- and C-terminal import and ratio between N- and C-terminal export rates) of all proteins tested, with their import and export kinetics compared. Error bars are standard error of the mean. **j**, Schematics for our proposed mechanism for orientation-dependent protein export from the nucleus. Similar to import, the nuclear export rate increases when the protein cargo is exported from its mechanically weaker terminus.

Having uncovered the protein orientation dependency for nuclear import and to get a holistic view on its effect on the bidirectional nucleocytoplasmic transport, we questioned whether the export process is also sensitive to the local structure and mechanical stability of the exporting protein cargoes. To this purpose, we engineered two additional constructs, based on the light-inducible nuclear localization signals (LINuS) optogenetic approach[43], NES-X-mCherry-LOV2-NLS, specifically tailored to measure protein export. In this case, the LOV2 protein is used to photocage an NLS

sequence to induce nuclear import on blue light exposure. Subsequently, when the light is turned off, the NLS is redocked and the protein is exported through an NES sequence and accumulates exponentially back into the cytoplasm (Fig. 5a). These constructs enabled us to measure the nuclear export rate for the protein cargo X from both its N-terminus (NES-X-mCherry-LOV2-NLS, export-forward construct) and its C-terminus (LOV2-NLS-mCherry-X-NES export-reverse construct) (Fig. 5b and Extended Data Fig. 7). In the absence of the cargo (X), the export dynamics of the mCherry protein alone are measured. These

light-induced export experiments globally reproduced the findings for import, that is, exporting from its N-terminus is faster than exporting it from its C-terminus (Fig. 5c–e and Supplementary Fig. 17). However, the difference in the export rates when exporting from both protein termini is significantly reduced when comparing them with their import counterparts (an asymmetry in the export rate constants of 26% versus 56% for import). This trend is maintained when testing the export rates for Ig27 (Fig. 5f–h and Supplementary Fig. 17) and for all the other model tested cargoes (X = protein L, spectrin R16, Spy0128 and its E117A and E258A mutants; Fig. 5i and Supplementary Figs. 18–22). Combined, these experiments conclude that export is also sensitive to the local mechanical stability of the exported protein, although to a much lesser degree than import (Fig. 5j).

The next step was to investigate the molecular mechanism by which the NPC senses unstructured regions of translocating proteins, especially in light of the absence of described ATPases that help pulling the shuttling protein cargo and the absence of preferential binding to nuclear transport receptors (Fig. 3b,c). We turned our attention to FG-Nups, which are collectively responsible for creating an efficient and selective barrier for cargo translocation[12,44,45]. We previously found[22] that Nup153 was able to specifically recognize unstructured, mechanically weak protein cargoes in large polyprotein constructs. We, hence, speculated that Nup153 might also be able to recognize localized unstructured protein regions within a single protein, potentially resulting from the partial and transient unfolding of the small protein cargoes as they enter the pore from their more labile terminus. To test this hypothesis, we compared, using the import-forward and import-reverse optogenetic constructs, the kinetics of nuclear accumulation of mCherry, Ig27 and Spy0128 E117A when imported from the N- and C-termini on silencing Nup153 with small interfering RNA (siNUP153; Supplementary Fig. 23). In all cases, we found that the effect that protein orientation has on their rate of nuclear passage and their overall nuclear accumulation when imported from its N- or C-termini was severely reduced when Nup153 was downregulated (Extended Data Fig. 8 and Supplementary Fig. 24).

The overarching conclusions that emerge from our experiments demonstrate that the localized mechanical stability determines the nuclear import (and to a lesser extent, export) rate of small, model proteins across the NPC. This resonates surprisingly well with the mechanistic description of protein translocation through several narrow pores of biological significance, such as mitochondria and proteasomes[4,8]. The potential biological significance of these findings,

however, remains unknown. Our observations with engineered model cargoes make it tempting to speculate that naturally occurring proteins that need to constantly shuttle in and out from the nucleus might regulate their nucleocytoplasmic transport according to their local nanomechanical properties.

To begin to explore this possibility, as a proof of principle, we focused on the SMAD4 TF, which modulates the transcription of transforming growth factor type β (TGFβ) genes, responsible for the regulation of cell growth, differentiation and apoptosis[46]. SMAD4 exhibits a well-structured C-terminus (with high β-strand content) and an α-helical rich N-terminus domain that contains the NLS sequence (Fig. 6a)[46,47]. Using smMT (Fig. 6b,e), we found that the C-terminus is an order-of-magnitude stiffer than the N-terminus (Fig. 6c,d,f,g). The marked structural and mechanical asymmetries of both termini makes SMAD4 a good candidate to probe whether its nuclear import rate is enhanced when it is imported to the nucleus through its mechanically soft terminus. To test this, we introduced an NES- and NLS-depleted version of the SMAD4 protein into the import-forward and import-reverse optogenetic constructs[48,49] (Fig. 6h). Using this approach, we measured a slower nuclear translocation when SMAD4 was imported from its mechanically stiffer C-terminus (Fig. 6i–k and Supplementary Fig. 25). Similar conclusions were drawn when using a NES-depleted (but keeping the endogenous NLS) SMAD variant in the import-forward and import-reverse constructs lacking the cMyc-NLS such that, under dark conditions, SMAD4 nuclear accumulation would be driven by its endogenous NLS (Extended Data Fig. 9). In agreement with the previous observations using model mechanical proteins, these results suggest that at least for the SMAD4 TF, naturally occurring proteins also exhibit a faster nuclear accumulation when imported from their mechanically weaker terminus, which contains the NLS in the case of SMAD4.

To test the statistical significance of these results across the whole pool of human TFs, we conducted a survey on the Human TFome[50] (containing 1,564 TFs). Using a predictor for NLS sequence recognition (cNLS)[51], we first identified the position of the NLS along the amino acid sequence of each TF, and then used a predictor of protein disorder (Metapredict[52]), to estimate the degree of disorder from the NLS position to the closest protein terminus (Fig. 6l and Supplementary Fig. 26). Strikingly, we found that although the NLS shows no preferred position along the TF sequence (Fig. 6m), the regions of the TF from the NLS sequence to the closest terminus tend to exhibit a significantly higher degree of disorder than the average disorder of that protein (Fig. 6n). One can, therefore, speculate that

**Fig. 6 | Human TFs preferentially enter the nucleus via mechanically and structurally labile regions. a**, Structural representation of SMAD4, with the N-terminal MH1 domain coloured blue and the C-terminal MH2 domain in green. The N-terminal NLS is highlighted in magenta. **b**, Schematics of the smMT construct for characterizing the mechanical stability of the MH1 N-terminus domain of SMAD4. **c**, Representative magnetic tweezers trajectory in a force ramp (1 pN s$^{-1}$), showing that the MH1 domain is mechanically vulnerable, unfolding at ~3.5 pN. **d**, Histogram showing the distribution of unfolding forces for the MH1 N-terminus domain of SMAD4. Data from $n = 175$ events measured from three individual molecules. **e**, Experimental single-molecule approach to test the nanomechanical properties of the MH2 C-terminus domain of SMAD4. **f**, The SMAD4 MH2 C-terminus domain is mechanically resistant, unfolding through multiple events occurring at forces greater than 40 pN. **g**, Histogram showing the distribution of unfolding forces for the MH2 C-terminus domain of SMAD4. Data from $n = 45$ events measured from three molecules. **h**, Schematics of the optogenetics constructs designed to monitor the nuclear import kinetics of SMAD4 when translocating from its mechanically weak MH1 N-terminal domain (left) or from its mechanically strong MH2 C-terminal domain (right). In both optogenetic constructs, the native NES and NLS sequences have been inactivated. **i–k**, Nuclear import kinetics of SMAD4 when translocating from its N- or C-terminus. **i**, Representative confocal images of U2OS cells after 30 min of the recovery phase. Scale bar, 10 μm. **j**, Average time courses of the nucleus-to-cytoplasm localization of the import-forward and import-reverse

optogenetic constructs comparing the nuclear import dynamics when SMAD4 is imported from its MH1 or MH2 terminus. **k**, Relative nuclear accumulation (left) and import rate calculated from fits to the recovery time courses. Bars indicate mean. Accumulation: N-term $K_e = 1.31 \pm 0.04$; C-term $K_e = 1.16 \pm 0.03$. Import rate: N-term $k_i = 1.05 \pm 0.04$ ks$^{-1}$; C-term $k_i = 0.62 \pm 0.02$ ks$^{-1}$. Significance levels for two-tailed Mann–Whitney non-parametric test. Accumulation, $P = 1.5 \times 10^{-2}$; import rate, $P = 7.8 \times 10^{-16}$. $n = 73$ (N-term); $n = 105$ (C-term) from $N > 3$ independent experiments. **l**, Disorder score shown for six example mechanosensitive TFs (1 indicating fully disordered and 0 indicating fully structured), highlighting the position of their NLS (magenta regions). The TF region encompassing from the NLS position to the closest terminus (blue regions) shows a significantly higher degree of disorder than the rest of the TF. **m**, Position of the NLS identified in 300 TFs. Box plots indicate the interquartile range, with the central line representing the median and whiskers extending to the minimum and maximum values. **n**, Integrated disorder score for the entire TF (grey), compared with the disorder score of the region comprising the NLS position to the closest terminus (blue). The TF region from the NLS position to the closest terminus is significantly more disordered than the entire TF sequence. Box plots indicate the interquartile range, with the central line representing the median and whiskers extending to the minimum and maximum values. Significance levels from two-tailed Mann–Whitney non-parametric test. $P = 8.8 \times 10^{-3}$. **o**, Schematics of our proposed model of TF import into the nucleus, where the local structural and mechanical properties of the cargo regulate the nuclear import dynamics.

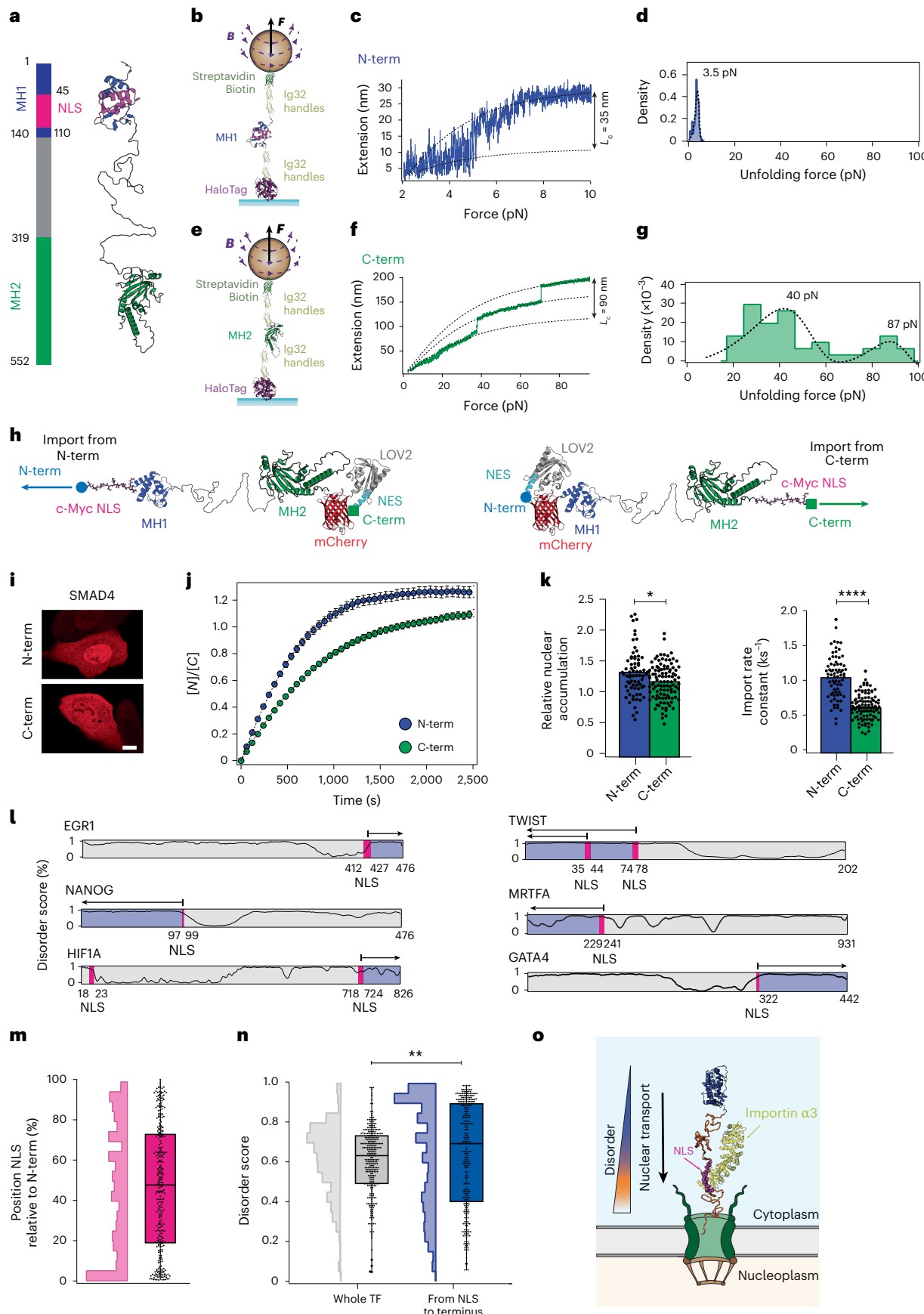

the general TF enrichment in disordered and flexible regions, in addition to contributing to enhancing binding with several promoters in the nucleus[53,54], might also facilitate TF binding to Nups, accelerating their nuclear import (Fig. 6o). Importantly, such increased disorder is independent of the NLS proximity to the terminus, suggesting that it is an intrinsic property of NLS-containing regions, and not of terminal ones (Supplementary Fig. 27). The localization of these loosely structured regions around the NLS sequence might underpin evolutionary pressure to import the protein cargo from the specific protein terminus that exhibits the least opposition to mechanical unfolding, to maximize its nuclear import efficiency.

Combined, the emerging picture arising from our experiments suggests that the NPC, and being admittedly larger and structurally more complex than other biological pores that require the transient unfolding of proteins, shares unanticipated transport similarities with them[8]. For example, in mitochondria, precursor proteins actively unfold to thread into the pore channels, the local structure of the mature domain adjacent to the targeting sequence being a determinant factor in the effectiveness and rate of the unfolding mechanism[55]. In fact, in vitro experiments in which mitochondrial-targeting sequences were specifically attached to the N-terminus and C-terminus of the same model Ig27 protein used here revealed that the import rate of the protein when imported from each terminus was different[3], and that, in agreement with our findings, the partial unfolding of the N-terminus is easier than the less-frequent unfolding of the C-terminal strands. Conceptually similar, biochemical experiments on purified archaeal and mammalian cell proteasomes elegantly showed that poorly structured domains generally exhibit enhanced degradation, and that the relative stability of each protein terminus determines the overall direction of protein translocation and proteasome degradation[1,9]. Related single-molecule experiments on the bacterial ClpXP and ClpAP AAA+ proteolytic machinery demonstrated the need for an unstructured region of the protein substrate to initially engage with the hexameric pore, and that this is achieved by the mechanical denaturation of the protein substrate fuelled by ATP hydrolysis[28,56,57]. In particular, the addition of the degron-targeting sequence to the N-terminus of the titin Ig27 protein results in a much faster degradation than when the protein is degraded from the C-terminus[58], in full agreement with our translocation experiments across the NPC (Fig. 2). Those experiments also concluded, using mutagenesis, that the local stability of proteins determines their degradation direction, and suggested that the placement of degron signals might have evolved to minimize the energetic cost of degradation[58]. Such a function-related consequence of the structural anisotropy of proteins was also observed in the VAT AAA+ unfoldase from *Thermoplasma acidophhilum*, demonstrating that intrinsically unstructured N- or C-termini directly initiate translocation[2]. Along these lines and from a more applied perspective, co-translocational unfolding also depends on the direction of pulling in artificial nanopores[25].

Within this broad context, our results suggest that the local properties of the protein cargo play an important role in the regulation of traffic across the NPC, affecting both import and export directions. We propose that the regulation of nuclear import kinetics is governed by both specific binding strength of the NLS sequence to the importin transporters and the local mechanical and structural properties of the protein cargo[22,59]. Of note, the local mechanical stability directly correlates with the level of structural disorder (Extended Data Fig. 10), implying that increasingly unstructured protein termini are mechanically more vulnerable. Mechanistically, partial mechanical unfolding of the mechanically weaker protein terminus close to the NLS signal is required to rapidly expose previously cryptic hydrophobic residues (Extended Data Fig. 10) that exhibit higher affinity with FG-Nups (ref. 59), specifically Nup153 (ref. 22), which preferentially binds to unstructured cargo regions. We speculate that the reported spring-like behaviour of Nup153 (ref. 60), probably facilitated by

Nup153's ability to 'dissolve' the dense FG-nup meshwork, assists the delivery of importin-cargo complexes to the nucleus. Whether a similar Nup-mediated mechanism of recognition of the mechanical vulnerability of proteins is at play during protein export remains to be investigated. Noteworthily, we still lack a precise quantification of the forces required to trigger partial protein unfolding inside the cell, which directly depend on the loading rate at which they would be pulled when threaded through the NPC. We speculate, however, that such forces will be much lower than those measured in our single-molecule nanomechanical experiments (and certainly in our SMD simulations; Supplementary Fig. 28). Single-molecule techniques generally use high pulling rates (for example, ~$10^3$ pN s$^{-1}$ in atomic force microscopy) that result in increased unfolding forces. However, recently, the loading rates of individual extracellular integrin tethers were measured to be between 0.5 and 5 pN s$^{-1}$ (ref. 61). Although the pulling rate for proteins translocating through the NPC remains experimentally inaccessible, it is likely that intracellular pulling rates are much lower than those used in both in vitro and in silico experiments, overall resulting in lower unfolding forces. Additionally, protein unfolding is accelerated at higher temperatures[62], which would further decrease the unfolding forces at 37 °C compared with those measured at room temperature in in vitro experiments. Our single-cell optogenetic experiments show the same hierarchy of mechanical stabilities measured using different techniques (Supplementary Fig. 28), in line with previous findings[21].

In summary, these experiments lend further evidence to the importance of the physicochemical properties of the protein cargo in the regulation of nucleocytoplasmic transport dynamics. Specifically, our findings provide sub-molecular insights into how the protein's local structure and nanomechanics affect its nuclear transport. Finally, it complements emergent research showing that mechanical forces converge at the NPC[14–16,63–66], in this case by demonstrating the NPC's sensitivity to subtle variations in the mechanical stability of specific regions within an individual translocating protein.

From an applied perspective, the precise regulation of the local stability of proteins might be potentially harnessed as a biotechnological tool to control and fine-tune the dynamics of specific protein cargo delivery to the cell nucleus.

## Online content

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

## Methods

### Plasmid construction

**LOV2 import-forward plasmid.** For the optogenetic experiments using the import-forward plasmid, the original pDN122 plasmid (Addgene, 72655)[29] was modified to remove the BglII site and the multicloning site. A cassette that contained an NheI-NLS-BamHI-KpnI-EheI sequence was then inserted between the NheI and AjiI sites present in the original pDN122 vector before mCherry. This strategy enables any DNA sequence to be swapped into the vector between the BamHI-KpnI sites. This is referred to as the import-forward cassette. We also designed an 'empty' construct to test the protein mCherry without the additional residues from restriction enzyme sites present in the cassette, giving NLS-mCherry-LOV2-NES, synthesized by GeneArt (Thermo Fisher Scientific), and then inserted into the pDN122 plasmid between the NheI and the XhoI sites. Cloning was achieved using FastDigest Enzymes (Thermo Fisher Scientific) and T4 DNA ligase (New England Biolabs).

**LOV2 import-reverse plasmid.** A DNA fragment containing the sequence NheI-LOV2-NES-mCherry-KpnI-BamHI-NLS-XhoI was synthesized by GeneArt (Thermo Fisher Scientific) and inserted into the NheI and XhoI sites of the modified pDN122 plasmid described above. The domains are, therefore, in reverse order compared with those in the pDN122 import-forward plasmid. This is referred to as the import-reverse cassette. The protein domain of interest was then inserted between the mCherry and NLS domains, using the restriction sites KpnI and BamHI. Note that the restriction enzyme sites have been swapped in this case so that the same restriction enzyme site links the NLS and protein of interest as in the import-forward cassette. We also designed an empty import-reverse construct to test the import of the C-terminus of the protein mCherry without the additional residues from restriction enzyme sites present in the cassette. A fragment containing NheI-LOV2-NES-mCherry-NLS-XhoI was synthesized by GeneArt and inserted into the NheI and XhoI sites of the pDN122 plasmid.

**SMAD4 plasmids.** To design the SMAD4 plasmids, two versions of SMAD4 were used. In both, the native NES sequence was inactivated by adding L146A and L148A mutations (ref. [49]). DNA fragments containing NheI-SMAD4 (NES inactive)-KpnI were synthesized by GeneArt and inserted into the pDN122 plasmid between the NheI and KpnI sites for the native SMAD4 import-forward construct, replacing both NLS and protein of interest. For the import-reverse plasmid, the same was done but with a KpnI-SMAD4 (NES inactive)-XhoI fragment synthesized by GeneArt. To design the SMAD4-cMyc-NLS plasmids, the SMAD4 (NES-inactive) template was used and the native SMAD4 NLS sequence, KKLKEKK, was modified to inactivate it by adding K45A/K46A/K48A/K50A/K51A (ref. [48]) mutations. Then, cMyc-NLS was added to either the N- or C-terminus of the protein. This was inserted into the pDN122 plasmids at the NheI and KpnI sites for the import-forward construct and at the KpnI and XhoI sites for the import-reverse version.

**NLS variant plasmids.** For the plasmids with different NLS sequences to the cMyc plasmid used throughout, the import-forward plasmid NLS-Ig27-mCherry-LOV2-NES and import-reverse plasmid LOV2-NES-mCherry-Ig27-NLS were used as templates. NLS sequences were then modified using site-directed mutagenesis to generate plasmids with the alternate NLS sequences, as listed in the table below.

| NLS name | Amino acid sequence |
|---|---|
| cMyc NLS (import NLS) | AAAKRVKLD |
| NES (photocaged for import constructs) | LKELADLNLD |
| IkBα NES (export NES) | PSTRIQQQLGQLTLENLQ |
| NLS (photocaged for export constructs) | KRLPDANLAAAAKKKKLD |
| High-affinity SV40 NLS | GPKKKRKV |
| Medium-affinity SV40 NLS | GPKKKAKV |
| Low-affinity SV40 NLS | GPKAKRKV |

**LOV2 export-forward plasmid.** For the optogenetic experiments using the export-forward measure protein export, the original pDN34 plasmid (Addgene, 61343)[43], REF was modified to remove the multicloning site. A cassette that contained an NheI-NES-BamHI-KpnI sequence was inserted into the modified pDN34 between the NheI and KpnI sites to create the export-forward cassette, allowing any DNA sequence to be swapped into the vector using the BamHI-KpnI sites. We also designed an empty construct to test the protein mCherry without additional residues from restriction enzyme sites present in the cassette, giving NES-mCherry-LOV2-NLS; this was synthesized by GeneArt (Thermo Fisher Scientific), and then inserted into the pDN34 plasmid between the NheI and XhoI sites. Cloning was achieved using FastDigest Enzymes (Thermo Fisher Scientific) and T4 DNA ligase (New England Biolabs).

**LOV2 export-reverse plasmid.** A DNA fragment containing the NheI-LOV2-NLS-mCherry-KpnI-BamHI-NLS-XhoI domains was synthesized by GeneArt (Thermo Fisher Scientific) and inserted into the NheI and XhoI sites of the modified pDN34 plasmid described above. The domains are, therefore, in reverse order compared with those in the pDN34 export-forward plasmid. The protein domain of interest was then inserted between the mCherry and NLS domains, using the restriction enzyme sites KpnI and BamHI. This is referred to as the export-reverse cassette. Note that the restriction enzyme sites have been swapped in this case so that the same restriction enzyme site links the NES and protein of interest as in the export-forward cassette. We also designed an empty export-reverse construct to test the export of the C-terminus of the protein mCherry without the additional residues from restriction enzyme sites present in the cassette. A fragment containing NheI-LOV2-NLS-mCherry-NES-XhoI was synthesized by GeneArt and inserted into the NheI and XhoI sites of the pDN34 plasmid.

### Protein expression and purification

**For magnetic tweezers experiments.** The modified pFN18A-HaloTag-Biotin plasmid (Addgene plasmid number 206039) was used using the BspEI and NheI sites to insert the monomer of interest (mCherry, circ-p mCherry, Ig27, R16, protein L, Smad4 MH1 domain and Smad4 MH2 domain). Plasmids were transformed into T7 competent cells (New England Biolabs), which carried a BirA plasmid for in vivo Biotin labelling of the C-term AviTag. Competent cells were grown in Luria–Bertamin medium supplemented with 100 mg ml$^{-1}$ of ampicillin at 37 °C. After reaching an OD$_{600}$ of ~0.6, cultures were induced with 0.5 mM of IPTG and 50 μM of Biotin (Invitrogen) and grown at 20 °C for 16 h. Cells were then resuspended in 20 mM of HEPES at pH 7.5 with 300 mM of NaCl, supplemented with 100 ml ml$^{-1}$ of lysozyme, 5 μg ml$^{-1}$ of RNase, 4 mM of PMSF, 4 mM of MgCl$_2$ and incubated on ice for 30 min. The lysate was passed through a French press (G. Heinemann), then incubated with HisPur cobalt resin (Thermo Fisher Scientific) and the His-tagged protein was eluted. This procedure was followed by gel filtration using a Superdex 200 10/300 GL column (Cytiva). Proteins were stored in a gel filtration buffer comprising 20 mM of HELPES (pH 7.5), 300 mM of NaCl and 10% glycerol at −80 °C immediately after gel filtration.

**For importin binding experiments.** Cells expressing 6His-Importin α3, NLS-I27-StreptagII and StreptagII-I27-NLS were harvested by centrifugation at 3,000$g$ for 20 min. The bacterial pellet from 500 ml of culture was resuspended in 25 ml of lysis buffer containing 20 mM of HEPES (pH 7.5), 300 mM of NaCl and 0.5 mM of TCEP in the presence of protease inhibitor cocktail tablets (Roche) and supplemented with 0.8 mg ml$^{-1}$ of lysozyme, 8 μg ml$^{-1}$ of DNase, 8 μg ml$^{-1}$ of RNase, 4 mM of PMSF and 10 mM of MgCl$_2$. After incubation on ice for 30 min, the cells were disrupted using a French press (G. Heinemann), and the lysate was cleared by centrifugation at 39,000$g$ for 45 min. The supernatant containing the soluble His-tagged protein was filtered with a 40-μm cell strainer and then loaded in a 1-ml StrepTrap XT column (for the strep-tagged proteins, or HisTrap HP for the 6His-Importin α3) that had

been previously equilibrated in a buffer containing 20 mM of HEPES at pH 7.5 and 300 mM of NaCl supplemented with 0.5 mM of TCEP (buffer A). The column was washed with 50 column volumes of buffer A. Strep-tagged protein elution was performed with buffer A containing 7 mM of Biotin. For the His-tagged protein, elution was performed using a step gradient with an AKTA device, using buffer A supplemented with 500 mM of imidazole as the elution buffer.

The eluted fractions were collected and analysed using sodium dodecyl sulfate–polyacrylamide gel electrophoresis (SDS-PAGE). Fractions containing the target protein were then pooled and dialysed overnight against buffer A. The protein sample was concentrated using a Vivaspin centrifugal device (Sartorious) and further purified on a Superdex 200 increase 10/300 GL column (Cytiva) using buffer A as the running buffer. Elutions were analysed using SDS-PAGE, and pure protein samples were dialysed ON again in buffer A before ITC experiments.

Cells co-expressing 6His-Importin α3 and strep-tagged NLS-cargo proteins were used for in vivo pull-down experiments. After resuspension of the 250-ml cell culture pellet in 25 ml of lysis buffer, cells were disrupted using a French press (G. Heinemann), followed by centrifugation at 39,000g for 45 min. Lysate was filtered with a 40-µm cell strainer and then loaded in a 1-ml StrepTrap XT column previously equilibrated in buffer A. After washing with 20 column volumes of buffer A, complexes were eluted with a solution containing buffer A supplemented with 7 mM of Biotin. The presence of 6His-Importin α3 in the eluted samples was confirmed by SDS-PAGE and western blotting.

## ITC

ITC measurements were performed on a MicroCal PEAQ-ITC calorimeter (Malvern Panalytical). Samples were dialysed overnight at 4 °C in 20 mM of HEPES (pH 7.5), 300 mM of NaCl and 0.5 mM of TCEP. Protein concentrations after dialysis were estimated by A280 absorbance using a V-760 spectrophotometer (Jasco).

Titrations were performed at 20 °C with 17 µM of Importin α3 in the cell and 170 µM of NLS-I27-Streptag, or Streptag-I27-NLS, in the syringe. The experiments were performed in duplicate. Data were analysed using the MicroCal PEAQ-ITC analysis software supplied by the manufacturer using nonlinear regression with the 'one set of sites' model. The thermodynamic parameters are averages from two experiments and are reported with standard deviation from the mean.

## Cell culture, transfection, protein knock-down and drug treatments

U2OS (American Type Culture Collection), NIH 3T3 (Francis Crick Institute) and HeLa (Francis Crick Institute) cells were grown in Dulbecco's modified Eagle's medium, high glucose, pyruvate (Merck), containing L-glutamine and supplemented with 10% fetal bovine serum (Merck) (for U2OS and HeLa cells) or 10% bovine calf serum (Merck) (for NIH 3T3 cells), 100 U ml$^{-1}$ of penicillin and 100 mg ml$^{-1}$ of streptomycin. For transient protein expression, cells were transfected with 1 µg of construct DNA using HD FuGENE (Promega) according to the manufacturer's instructions. For protein knock-down experiments, siRNA (non-targeting; D-001810-10-05, against Nup153; L-005283-00-0005) (Horizon Discovery) was transfected into cells with Lipofectamine RNAimax reagent (Invitrogen) using the manufacturer's protocol. Where indicated, U2OS cells were treated with 20 nM of leptomycin B (Sigma-Aldrich) after 10 min of blue light activation. Cells were then exposed to blue light for an additional 10 min to allow for leptomycin-mediated exportin inhibition before the blue light was turned off for the recovery phase of the experiment.

## Protein extraction and immunoblotting

Cells were lysed using radioimmunoprecipitation assay buffer with a protease inhibitor cocktail. Protein concentration of lysates was determined with Pierce BCA assay (Thermo Fisher Scientific). The lysate was separated by running on an SDS-PAGE gel and transferring it to a nitrocellulose membrane. Membranes were incubated in a blocking buffer (5% skimmed milk powder in TBS containing 0.2% Tween-20) for 1 h at room temperature and then incubated overnight with antibodies against Nup153 (A301-788A-T, Bethyl Laboratories), importin α3 (MAB8204, R&D Systems) and glyceraldehyde-3-phosphate dehydrogenase (ab8245, Abcam). Antibodies were visualized using the chemiluminescence SuperSignal West Pico PLUS detection system (Thermo Scientific). Densitometric analysis was performed using ImageJ2 (v2.16.0).

## Live-cell image acquisition

Before image acquisition, HEPES (pH 7) was added (50 mM) to the cells. Live-cell imaging was performed in an enclosed environmental chamber (37 °C) with a confocal Nikon A1R inverted microscope using a × 60 1.40-numerical-aperture oil-immersion objective. Up to six sample positions were recorded for at least three independent replicates. The microscope was operated with the Nikon Perfect Focus System and controlled by NIS-Elements software (v5.21.02).

For optogenetic experiments, image acquisition was performed throughout the activation and recovery phases (Fig. 1a). The activation of the optogenetic constructs was performed with a constant blue light (mean wavelength, 488 nm) illuminating the sample, followed by a recovery phase without blue light. Temporal parameters for imaging are indicated in the table below.

| Construct | Activation phase | Activation min/frame | Recovery phase | Recovery min/frame |
|---|---|---|---|---|
| **Standard parameters** | 10 min | 1 | 45 min | 1 |
| **NLS, low** | 10 min | 1 | 120 min | 3 |
| **Export (LiNUS)** | 20 min | 1 | 30 min | 1 |
| **Leptomycin treatments** | 20 min | 1 | 120 min | 3 |
| **SMAD4, native NLS** | 10 min | 1 | 90 min | 2 |

## Live-cell quantification

**Protein import.** Image quantification was performed using a custom MATLAB script[22]. Briefly, the collected images were background corrected, and then, the total fluorescence intensities of the nucleus and cell were defined manually. The nuclear and cytoplasmic signals were normalized by the total cell intensity, and these quantities were used as a proxy for the protein concentrations. Nuclear import kinetics were assumed to follow first-order kinetics as

$$[N](t) = [N]_e - ([N]_e - [N]_0)e^{-k_I t},$$

where $[N]_0$ and $[N]_e$ are the initial and steady-state nuclear concentrations, respectively, and $k$ is he effective import rate constant. Similarly, the cytoplasmatic concentration was assumed to evolve as

$$[C](t) = [C]_e + ([C]_0 - [C]_e)e^{-k_E t}.$$

Finally, introducing the relative nucleous-to-cytoplasm volume, which, assuming mass conservation, obeys $v = ([C]_0 - [C]_e)/([N]_e - [N]_e)$, the nucleus-to-cytoplasm concentration follows:

$$\frac{[N]}{[C]}(t) = \frac{K_e \left(1 - e^{-kt}\right)}{\left(1 + v K_e e^{-kt}\right)}, \tag{1}$$

where $K_e = k_I/k_I + k_E$ is the relative nucleus-to-cytoplasm accumulation in the steady state. Also, $k = k_I + k_E$, where $k_I$ and $k_E$ are the active import and export rate constants, respectively.

We quantified the live-cell images by measuring the nuclear and cytoplasmic signals and fitting the time course of $[N]/[C]$ during the recovery phase to equation (1) to extract the relative nuclear

accumulation $K_e$ and import rate $k_I$. Spurious cell measurements were removed from the analysis by applying a semiautomatic filtering protocol discarding unphysical fits and outliers and poorly activated cells (where the nucleus does not empty during the activation phase).

Protein export experiments were analysed following an analogous procedure, tracking the time course of [C]/[N], which follows:

$$\frac{[C]}{[N]}(t) = \frac{K_e \left(1 - e^{-kt}\right)}{\left(1 + \upsilon K_e e^{-kt}\right)}, \tag{2}$$

where $K_e = k_E/k_I + k_E$ is the relative cytoplasm-to-nucleus accumulation and $k = k_I + k_E$.

### Definition of percentage of split accumulation and percentage of split import/export rate

To quantify the level of N/C-term asymmetry, we defined the split observed between the N- versus C-term accumulation and import/export rates as

$$\text{Split} = 100 * \left(\frac{K_e^N}{K_e^C} - 1\right),$$

where $K_e^N$ is the average accumulation for the N-term forward construct and $K_e^C$, for the C-term reverse construct. An analogous expression was used for the import/export rates. Therefore, this quantity represents the degree of asymmetry, being positive if the N-term forward construct is faster or negative when the C-term reverse construct is faster.

### Magnetic tweezers experiments

The smMT experiments were conducted on a custom setup[67]. Briefly, the setup is built on an inverted microscope (Nikon), with the magnets (N52) mounted on a voice coil (Equipment Solutions) to control their vertical position and placed on top of a × 100 oil-immersion objective (Nikon) mounted on a piezo-electric actuator (PI). The illumination is performed by red light-emitting diode cold light source (Thorlabs), whereas the image acquisition is achieved using a complementary metal–oxide–semiconductor camera (Ximea). Control of the magnets position and the piezo is achieved using a multifunction data acquisition card (National Instruments), using custom data acquisition software.

The molecule of interest was tethered to a superparamagnetic Dynabead M-270 streptavidin-coated bead (Invitrogen), which resists forces up to 120 pN. The sample was prepared in custom fluid chambers consisting of two glass coverslips (Menzel-Glaser) separated by a laser-cut parafilm pattern. The fluid chamber was functionalized with HaloTag O4 ligand[67] to achieve covalent and specific anchoring of the N-terminus HaloTag protein constructs, and amino-coated non-magnetic beads were used as reference beads. The chamber is passivated using Tris blocking buffer (20 mM of Tris-HCL (pH 7.4), 150 mM of NaCl, 2 mM of MgCl$_2$ and 1% w/v sulfhydryl-blocked BSA).

The protein was incubated in the fluid chamber for ~30 min (1–5 nM) to ensure attachment, and experiments were conducted in PBS containing 10 mM of ascorbic acid (pH 7.4) to minimize oxidative damage. M-270 beads (~20 μl) were added in the chamber and incubated for 5 min before force application. To unfold the protein of interest, a linear force ramp at a loading rate of 1 pN s$^{-1}$ was applied.

To analyse the force-ramp experiments, the step sizes for each unfolding event were converted to contour length increment $\Delta L_C$ using the freely jointed chain model:

$$x(F) = \Delta L_C \left[ \coth\left(\frac{Fl_K}{kT}\right) - \frac{kT}{Fl_K} \right],$$

where $l_K$ is the Kuhn length, assumed to be of 1.1 nm, and $kT = 4.11$ pN nm. Each event was then characterized by its unfolding force and contour length increment, which were represented as a $\Delta L_C$ versus $F$ scatter plot.

### SMD simulations

The GROMACS 2018.4 (ref. 68) simulation package was used for all simulations, using the AMBER-ff99sb-ILDN (ref. 69) force field with Joung ions[70] and the Tip3p water model[71]. Before the simulations, the proteins were solvated, neutralized with the necessary ion concentration and the energy of the initial configuration was minimized using the steepest-descent algorithm. The protein was then subjected to 100-ps *NVT* and 100-ps *NPT* equilibration. The *NPT* ensemble was kept at 300 K using the velocity-rescale thermostat and the pressure was controlled at 1 atm using the Parrinello–Rahman barostat with a relaxation time of 2 ps (ref. 72).

SMD simulations for the mCherry (PDB: 6YLM) protein were performed using the N-terminal and C-terminal amino acids as pulling groups and using a constant-force protocol at a force of $F = 350$ pN. The extension of the protein was monitored every 1,000 time steps.

For the N-term and C-term pulling protocols on the Ig27 (PDB: 1TIT), protein L (1HZ6) or spectrin R16 (based on PDB 1U5P, residues: 1,763–1,872, and using MODELLER to model the human R16 structure), the pulling groups were the N-term amino acid and the remainder of the protein but for the N-terminal strand (N-term) or the C-terminal amino acid and the remainder of the protein but for the C-terminal strand (C-term). Therefore, this pulling mode correspondingly forced the N-term or C-term regions to unfold first and keeping the rest of the protein folded, mimicking a directional unfolding process. In both protocols, the pulling velocity was $\upsilon = 0.001$ nm ps$^{-1}$ and the spring constant $k = 1,000$ pN nm. Hydrogen bonds were analysed using PyMOL with a cut-off distance of 4 Å and an angle cut-off of 30°.

### Analysis of disorder in human TFs

The Human TFome database was surveyed[50] (comprising 1,564 TFs). On the basis of the consensus amino acid sequence of their UniProt sequence, the NLS sequence position was identified using the cNLS mapper[51] resource tool, which predicts NLS sequences, also providing a score based on its amino acid composition. Out of the 1,564 TFs, 305 TFs were identified. We only considered one NLS per TF (taking the one with the highest score), and for those TFs with multiple isoforms, we only analysed one of them, to prevent biases from structurally similar proteins. For these TFs, the disorder score for each amino acid was determined using Metapredict[52]. Then, the disorder score for (1) the full TF and (2) from the NLS to the closest terminus was determined by calculating the average disorder score for (1) the full TF sequence and (2) from the NLS to N-terminus if the NLS was located within the first 50% of the sequence, or from the NLS to C-terminus if it was located within the second 50% of the sequence. Supplementary Fig. 26 shows a schematic of the analysis pipeline.

### Reporting summary

Further information on research design is available in the Nature Portfolio Reporting Summary linked to this article.

## Data availability

The DNA and protein sequences used in this study are included as Supplementary Data 1. Raw data for the TFs analysis are included as Supplementary Data 2. All other data that support the plots within this paper and other findings of this study are available from the corresponding authors upon reasonable request. Source data are provided with this paper.

## Code availability

The code is available from the corresponding authors upon reasonable request.

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

## Acknowledgements

We thank F. Panagaki for help in the preliminary experiments. Via our membership of the UK's HEC Materials Chemistry Consortium, which is funded by EPSRC (EP/L000202), this work used the UK Materials and Molecular Modelling Hub for computational resources, MMM Hub, which is partially funded by EPSRC (EP/T022213/1, EP/W032260/1 and EP/P020194/1). This work was supported in part by the Francis Crick Institute that receives its core funding from Cancer Research UK (CC0102), the UK Medical Research Council (CC0102) and the Wellcome Trust (CC0102). Cell lines used in this work were provided by the Cell Science facility of the Francis Crick Institute. R.T.-R. was the recipient of a King's Prize Fellowship. J.G. is the recipient of a BBSRC LiDO doctoral studentship. This work is supported by the European Commission (Mechanocontrol, grant agreement number 731957), BBSRC sLoLa (BB/V003518/1), Leverhulme Trust Research Leadership Award (RL 2016-015), Wellcome Trust Investigator Award (212218/Z/18/Z) and Royal Society Wolfson Fellowship (RSWF/R3/183006) to S.G.-M.

## Author contributions

S.G.-M. conceived the research. R.T.-R. and S.G.-M. designed the research. N.M., C.E.-L. and J.G. conducted and analysed the cellular optogenetic experiments. R.T.-R. analysed the cellular translocation data. R.T.-R. conducted the single-molecule mechanical experiments and analysed the data. R.T.-R. conducted and analysed the SMD simulations and structural prediction datasets. P.P., B.L. and J.W. engineered the protein constructs for binding assays and for cellular experiments, and expressed and purified the protein constructs for single-molecule experiments. L.M. conducted and analysed the ITC binding assays. R.T.-R., N.M. and S.G.-M. wrote the paper. All authors contributed to revising and editing the manuscript.

## Funding

## Competing interests

The authors declare no competing interests.

## Additional information

**Extended data** is available for this paper at https://doi.org/10.1038/s41567-026-03242-2.

**Correspondence and requests for materials** should be addressed to Rafael Tapia-Rojo or Sergi Garcia-Manyes.

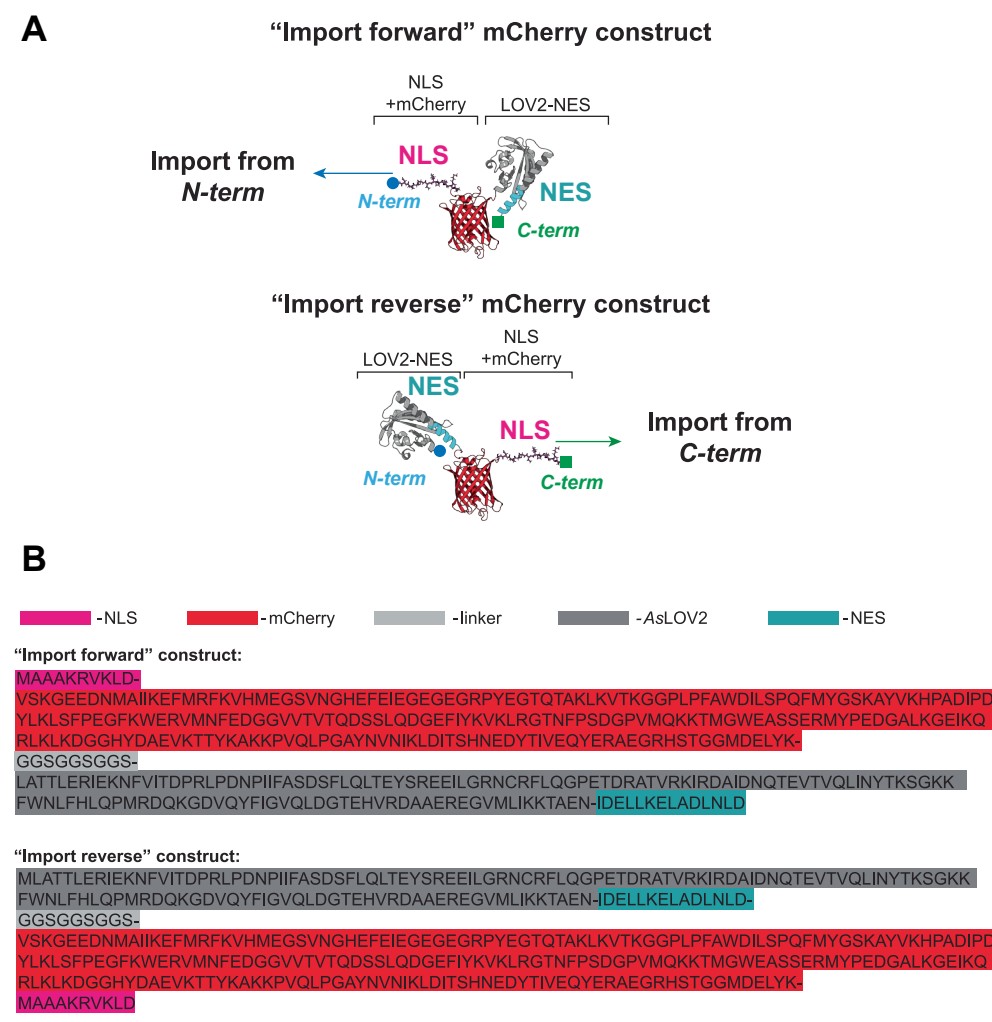

**Extended Data Fig. 1 | Molecular strategy for the "Import forward" and "Import reverse" mCherry construct. (A)** Schematics showing a structural model of the "Import forward" (upper) and "Import reverse" constructs (lower). **(B)** Amino acid sequence for the "Import forward" and "Import reverse" constructs, with color code indicating each of the molecular components.

## A

### "Import forward" construct

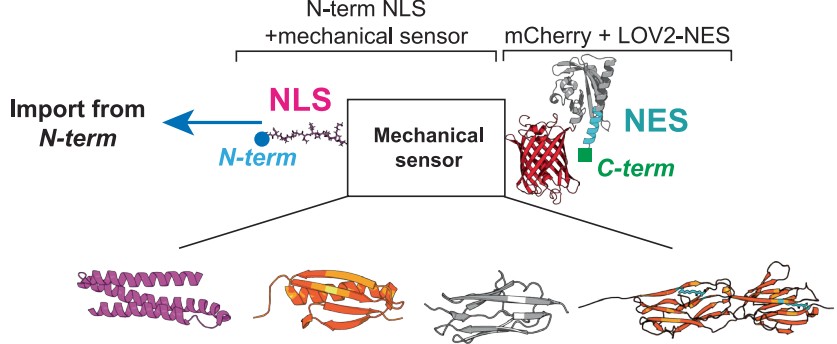

### "Import reverse" construct

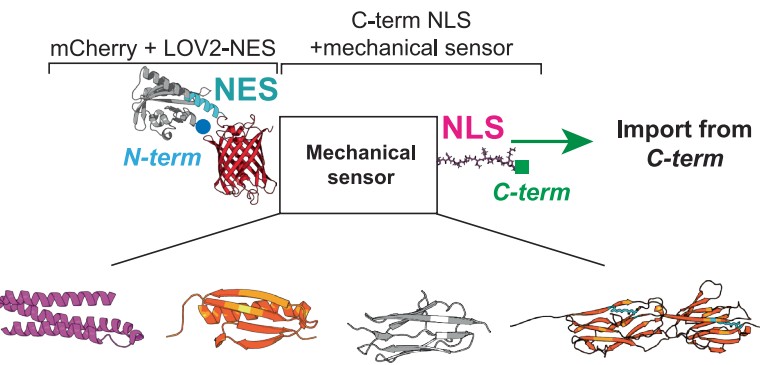

## B

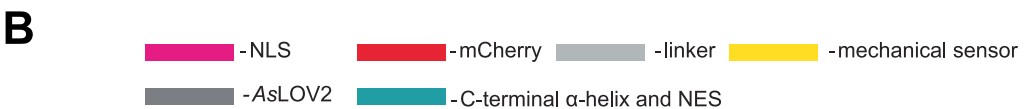

**"Import forward" construct:**

MAAAKRVKLD-
GS-**X**-GTGAG
VSKGEEDNMAIIKEFMRFKVHMEGSVNGHEFEIEGEGEGRPYEGTQTAKLKVTKGGPLPFAWDILSPQFMYGSKAYVKHPADIPD
YLKLSFPEGFKWERVMNFEDGGVVTVTQDSSLQDGEFIYKVKLRGTNFPSDGPVMQKKTMGWEASSERMYPEDGALKGEIKQ
RLKLKDGGHYDAEVKTTYKAKKPVQLPGAYNVNIKLDITSHNEDYTIVEQYERAEGRHSTGGMDELYK-
GGSGGSGGS-
LATTLERIEKNFVITDPRLPDNPIIFASDSFLQLTEYSREEILGRNCRFLQGPETDRATVRKIRDAIDNQTEVTVQLINYTKSGKKF
WNLFHLQPMRDQKGDVQYFIGVQLDGTEHVRDAAEREGVMLIKKTAEN-IDELLKELADLNLD

**"Import reverse" construct:**

MLATTLERIEKNFVITDPRLPDNPIIFASDSFLQLTEYSREEILGRNCRFLQGPETDRATVRKIRDAIDNQTEVTVQLINYTKSGKK
FWNLFHLQPMRDQKGDVQYFIGVQLDGTEHVRDAAEREGVMLIKKTAEN-IDELLKELADLNLD-
GGSGGSGGS-
VSKGEEDNMAIIKEFMRFKVHMEGSVNGHEFEIEGEGEGRPYEGTQTAKLKVTKGGPLPFAWDILSPQFMYGSKAYVKHPADIPD
YLKLSFPEGFKWERVMNFEDGGVVTVTQDSSLQDGEFIYKVKLRGTNFPSDGPVMQKKTMGWEASSERMYPEDGALKGEIKQ
RLKLKDGGHYDAEVKTTYKAKKPVQLPGAYNVNIKLDITSHNEDYTIVEQYERAEGRHSTGGMDELYK-
GAGGT-**X**-GS
MAAAKRVKLD

**Extended Data Fig. 2 | Molecular strategy for probing the effect of protein orientation on nuclear import kinetics inside cells. (A)** Schematics showing a structural model of the "Import forward" (upper) and "Import reverse" constructs (lower), highlighting the insertion site for the mechanical sensor of interest, from left to right R16, protein L, Ig27, and Spy0128. **(B)** Amino acid sequence for the "Import forward" and "Import reverse" constructs, with color code indicating each of the molecular components.

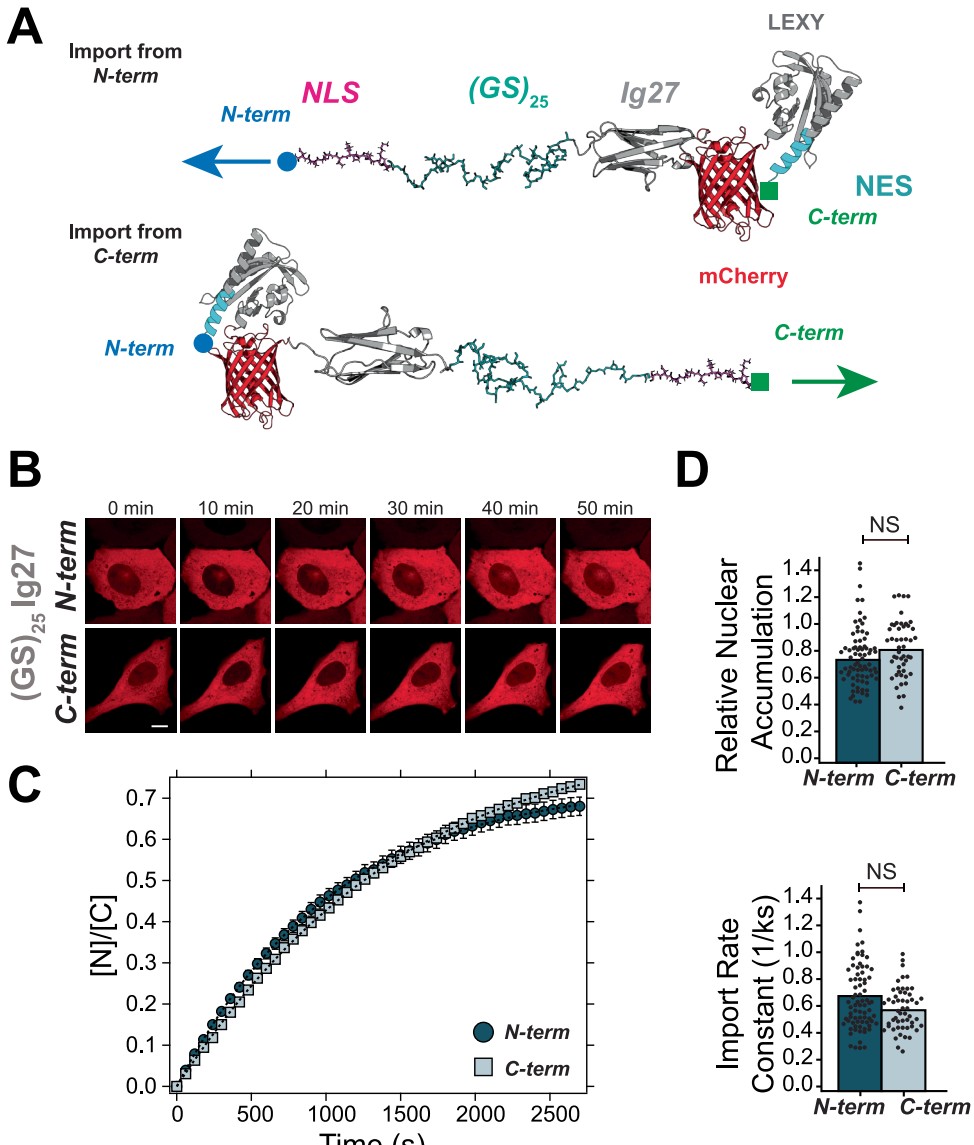

**Extended Data Fig. 3 | Adding an unstructured (GS)$_{25}$ tag to the N- or the C-terminus of Ig27 abrogates the directionality in its nuclear import.** (**A**) Structural model of the protein construct to probe the import kinetics of the (GS)$_{25}$-Ig27 "Import forward" (upper) and Ig27-(GS)$_{25}$ "Import reverse". (**B-D**) Nuclear import kinetics of Ig27 when translocating from the N- or C-terminus with a (GS)$_{25}$ tag adjacent to the NLS. (**B**) Representative confocal images of U2OS cells, scale bar 10 μm. (**C**) Average time-courses of the nucleus-to-cytoplasm localisation of the import "Import forward" and "Import reverse" (GS)$_{25}$-Ig27 or Ig27-(GS)$_{25}$. (**D**) Relative nuclear accumulation (upper) and import rate (lower) calculated from fits to the recovery time courses. Bars indicate mean ± SEM. Accumulation: N-term $K_e = 0.75 \pm 0.03$; C-term $K_e = 0.81 \pm 0.03$. Import rate: N-term $k_i = 0.66 \pm 0.03 \text{ ks}^{-1}$; C-term $k_i = 0.59 \pm 0.02 \text{ ks}^{-1}$. Significance levels for two-tailed Mann-Whitney non-parametric test. Accumulation, $P = 0.05$; Import rate, $P = 0.16$. $n = 77$ (N-term); $n = 53$ (C-term).

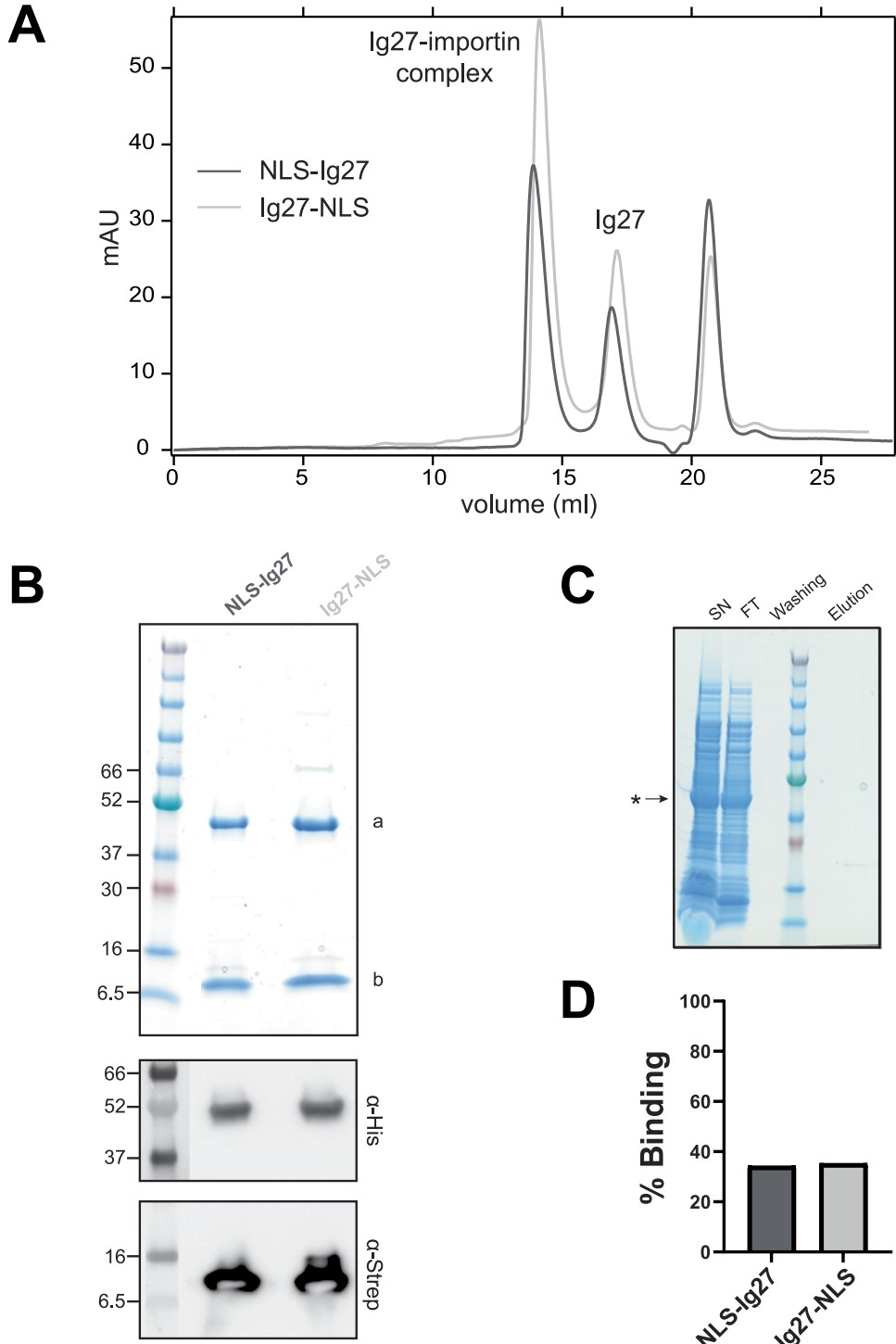

**Extended Data Fig. 4 | Importin α3 interacts with coexpressed NLS-Ig27 to a similar extent regardless of the N- or C-term orientation of the NLS.** (**A**) Gel filtration chromatograms of Importin α3-NLS cargo complexes performed using a Superdex 200 Increase 10/300 GL column. The retention volumes of the two prominent peaks correspond to the formation of stable complexes between Importin α3 and NLS-cargo proteins or the peaks corresponding to the free NLS-cargo proteins. (**B**) Complex formation in E-coli cells confirmed by SDS-PAGE. Two distinct bands, corresponding to 6His-Importin α3 (a) and Strep-tagged NLS-cargo (b), are clearly observed. The elution of 6His-Importin α3 after a pull-down using Strep-tagged NLS-cargo as bait confirms direct binding. Experiments were repeated at least twice with consistent results. Protein identities were confirmed using anti-His and anti-Strep antibodies. (**C**) Negative control showing the pull-down of 6His-Importin α3 (*) in the absence of the bait protein (Strep-tagged NLS-cargo). The absence of Importin α3 in the elution sample confirms the specificity of the interaction. (**D**) Quantification of the percentage of Importin α3 retention, calculated from band intensities measured with Fiji (ImageJ) from the SDS-PAGE gel. Values were normalised to the corresponding Strep-tagged NLS-cargo proteins. NLS-Ig27 = 34.30, Ig27-NLS = 35.32.

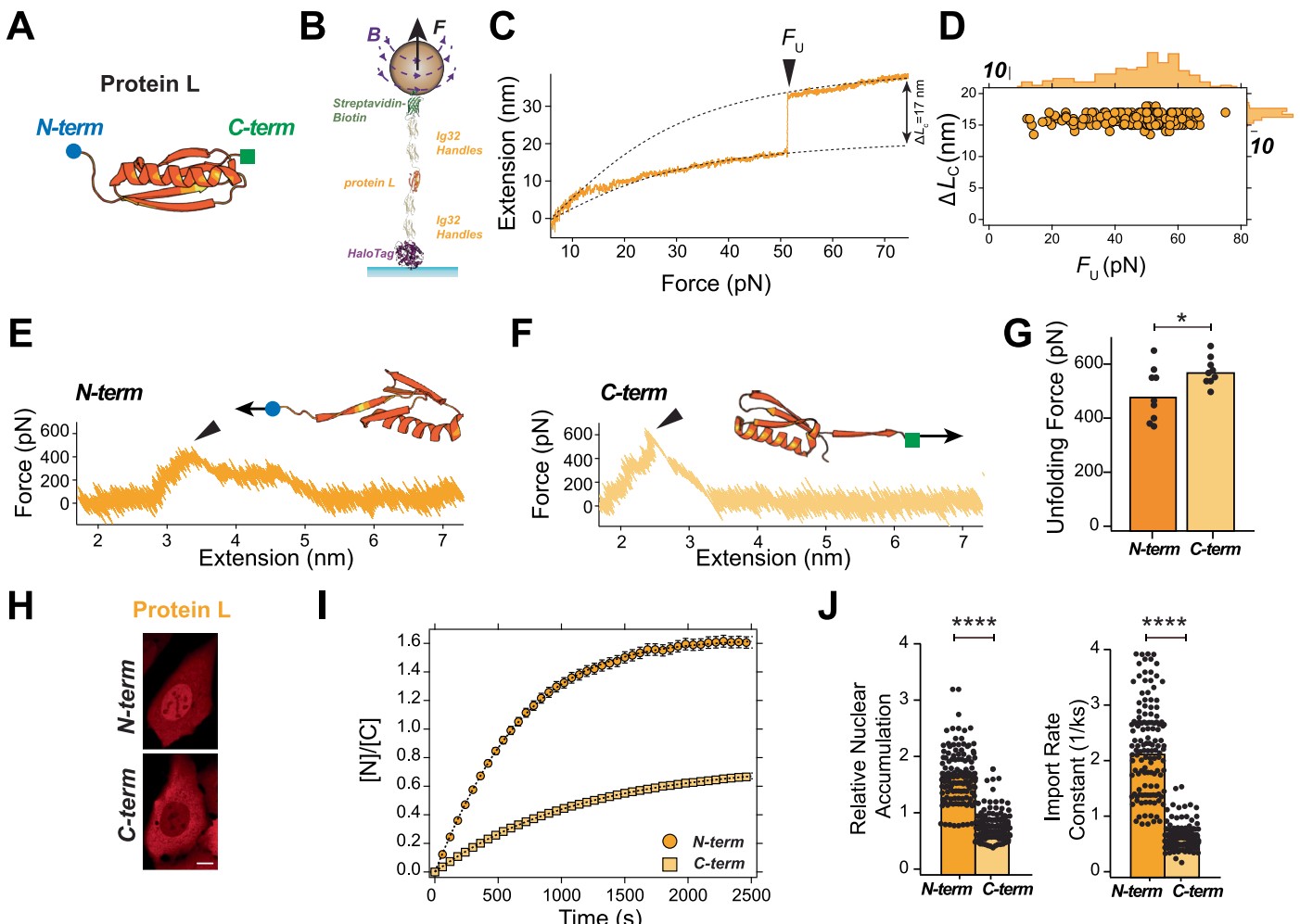

**Extended Data Fig. 5 | Asymmetric nuclear import in protein L due to its structural and mechanical anisotropy.** (**A**) Structure of protein L, with an α/β tertiary structure, where the N-terminus is marked in blue, and the C-terminus in green. (**B**) Schematics of the single-molecule magnetic tweezers experiment. (**C**) Representative magnetic tweezers trajectory in a force ramp (1 pN/s), showing that Protein L unfolds in a single event at a force of ~50 pN. (**D**) Scatter plot of the increase in contour length ($\Delta L_c$) versus unfolding force ($F_U$) for each unfolding event, together with the marginal probability distributions for $\Delta L_c$ and $F_U$. Data from $n = 150$ events measured from 10 molecules. (**E-F**) Force-extension trajectory from molecular dynamics simulation for the N-term (**E**) and C-term (**F**) pulling, showing the rupture event (arrow), where the N-term/C-term strand unfold. (**H**) Distribution of unfolding forces corresponding to unravelling each

individual terminus (N-term/C-term), obtained from 10 SMD simulations (pulling speed $v = 1$ nm/ns). Significance level from two-tailed T-test, $P = 1.2 \times 10^{-2}$. (**G-J**) Nuclear import kinetics of Protein L when translocating from its N- or C-terminus. (**G**) Representative confocal images of U2OS cells after 30 min, in the recovery phase, scale bar 10 μm. (**H**) Average time-courses of the nucleus-to-cytoplasm localisation of the "Import forward" and "Import reverse" optogenetic constructs containing the (X=Protein L) cargo. (**J**) Relative nuclear accumulation (left) and import rate calculated from fits to the recovery time courses. Bars indicate mean. Accumulation: N-term $K_e = 1.66 \pm 0.04$; C-term $K_e = 0.76 \pm 0.02$. Import rate: N-term $k_I = 2.26 \pm 0.07$ ks⁻¹; C-term $k_I = 0.64 \pm 0.02$ ks⁻¹. Significance levels for two-tailed Mann-Whitney non-parametric test. Accumulation, $P = 0$; Import rate, $P = 0$. $n = 129$ (N-term); $n = 116$ (C-term) obtained from $N > 3$ independent experiments.

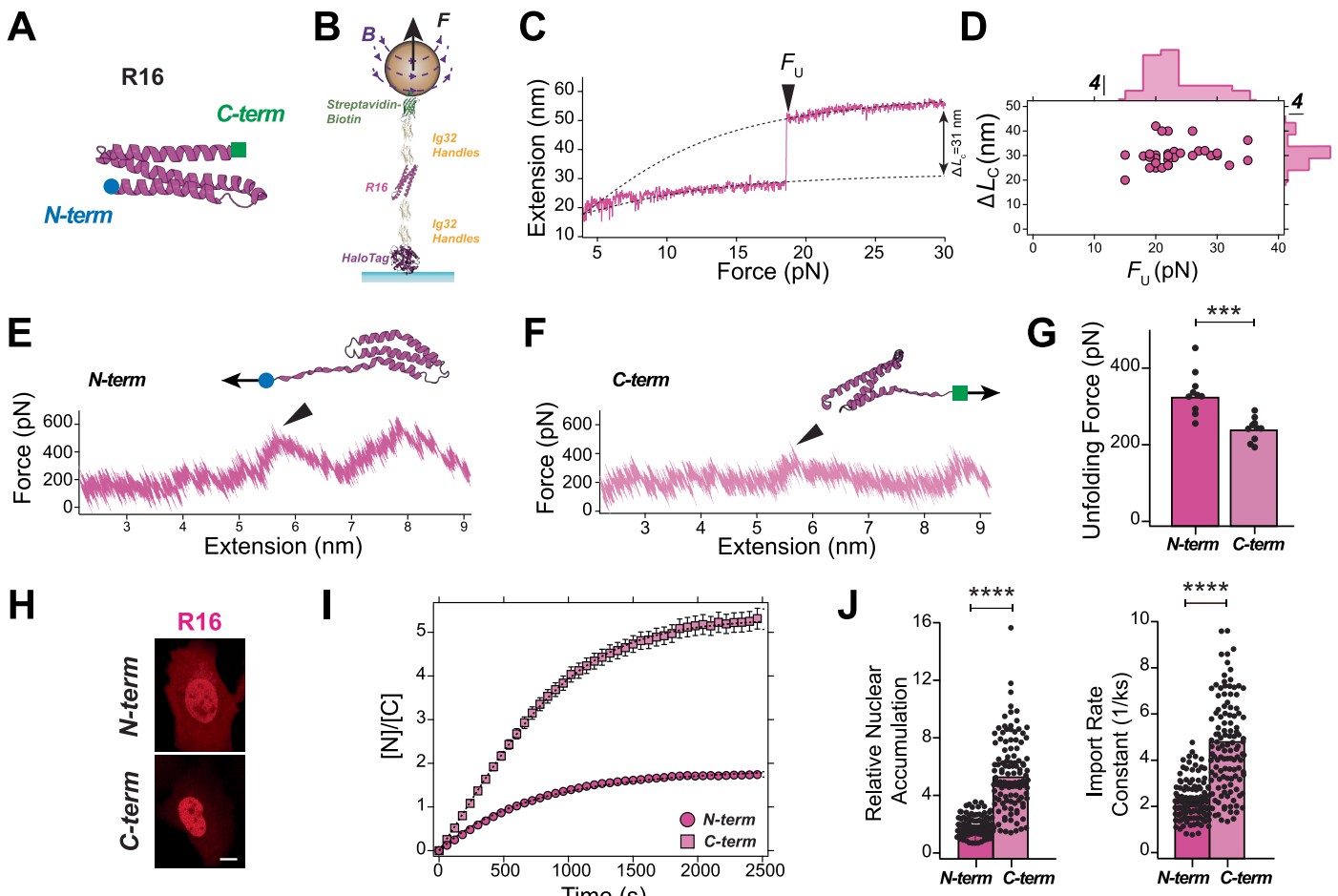

**Extended Data Fig. 6 | Asymmetric nuclear import in the spectrin R16 domain due to its mechanical and structural anisotropy.** (**A**) Structure of the R16 domain of spectrin, composed of a three α-helical bundle. The N-terminus is marked in blue, and the C-terminus in green. (**B**) Schematics of the single molecule magnetic tweezers construct to test the nanomechanical properties of R16. (**C**) R16 unfolds in a single event at a force as low as ~18 pN. (**D**) Scatter plot of the increase in contour length ($\Delta L_c$) versus unfolding force ($F_U$) for each unfolding event, together with the marginal probability distributions for $\Delta L_c$ and $F_U$. Data from $n = 35$ events measured from 8 molecules. (**E-F**) Force-extension trajectory obtained from molecular dynamics simulations where R16 is pulled from the N-term (**E**) and the C-term (**F**). (**G**) Distribution of unfolding forces corresponding to the independent unravelling of the N-term and C-term, obtained from 10 SMD simulations (pulling speed $v = 1$ nm/ns). Significance level from two-tailed T-test, $P = 1.7 \times 10^{-4}$. (**H-J**) Nuclear import kinetics of R16 when translocating from its N- or C-terminus. (**H**) Representative confocal images of U2OS cells after 30 min of the recovery phase, scale bar 10 μm. (**I**) Average time-courses of the nucleus-to-cytoplasm localisation of the "Import forward" and "Import reverse" optogenetic constructs containing R16. (**J**) Relative nuclear accumulation (left) and import rate (right) calculated from fits to the recovery time courses. Bars indicate mean. Accumulation: N-term $K_e = 1.81 \pm 0.06$; C-term $K_e = 5.46 \pm 0.24$. Import rate: N-term $k_1 = 2.31 \pm 0.08$ ks$^{-1}$; C-term $k_1 = 4.63 \pm 0.18$ ks$^{-1}$. $n = 110$ (N-term); $n = 119$ (C-term). Significance levels for two-tailed Mann-Whitney non-parametric test. Accumulation, $P = 0$; Import rate, $P = 5.6 \times 10^{-16}$. $n = 110$ (N-term); $n = 119$ (C-term) obtained from $N > 3$ independent experiments.

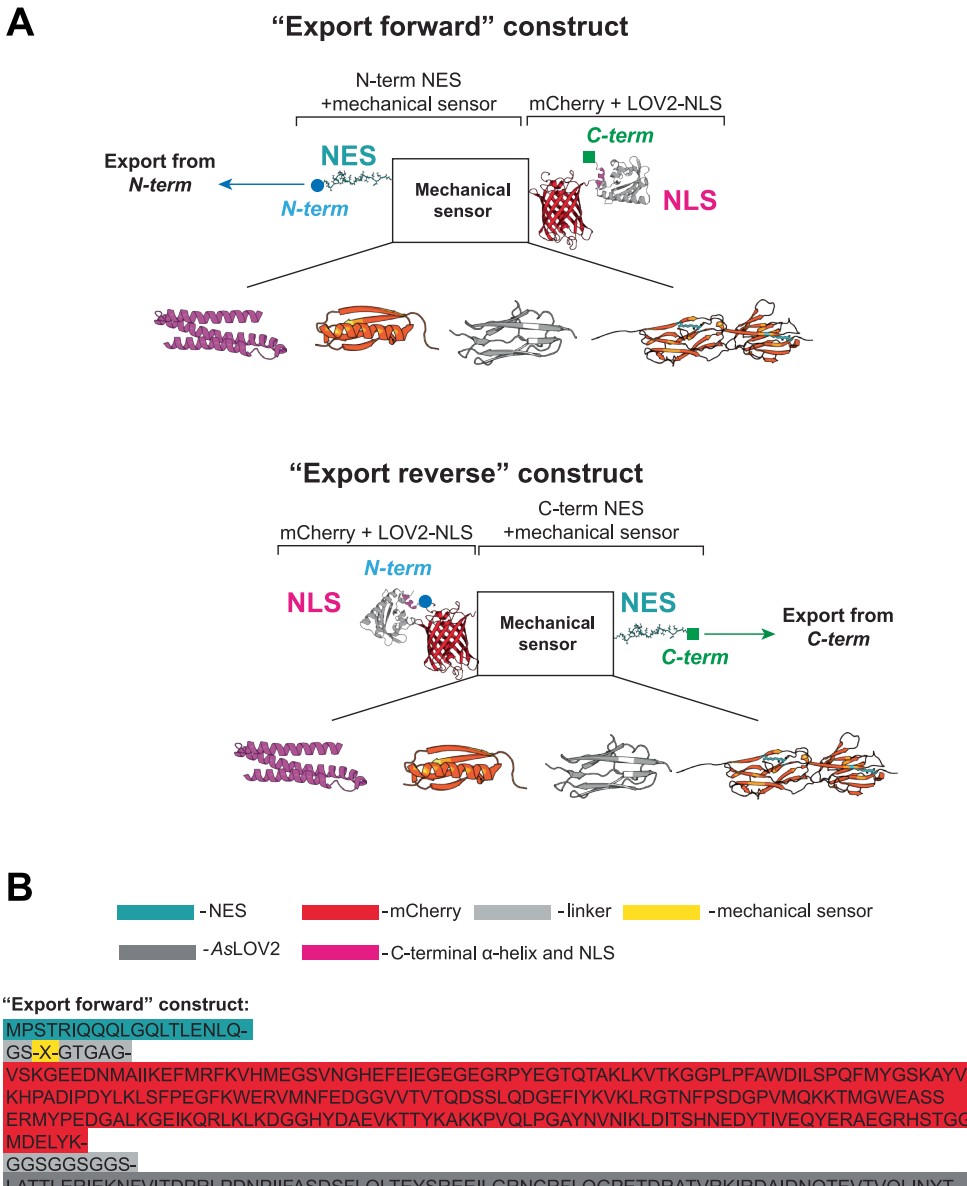

**A**

**"Export forward" construct**

**"Export reverse" construct**

**B**

- NES  - mCherry  - linker  - mechanical sensor
- *As*LOV2  - C-terminal α-helix and NLS

**"Export forward" construct:**

MPSTRIQQQLGQLTLENLQ-
GS-X-GTGAG-
VSKGEEDNMAIIKEFMRFKVHMEGSVNGHEFEIEGEGEGRPYEGTQTAKLKVTKGGPLPFAWDILSPQFMYGSKAYV
KHPADIPDYLKLSFPEGFKWERVMNFEDGGVVTVTQDSSLQDGEFIYKVKLRGTNFPSDGPVMQKKTMGWEASS
ERMYPEDGALKGEIKQRLKLKDGGHYDAEVKTTYKAKKPVQLPGAYNVNIKLDITSHNEDYTIVEQYERAEGRHSTGG
MDELYK-
GGSGGSGGS-
LATTLERIEKNFVITDPRLPDNPIIFASDSFLQLTEYSREEILGRNCRFLQGPETDRATVRKIRDAIDNQTEVTVQLINYT
KSGKKFWNLFHLQPMRDQKGDVQYFIGVQLDGTEHVRDAAEREGVMLIKKTAEN-IDEAAKRLPDANLAAAAKKKKLD

**"Export reverse" construct:**

MLATTLERIEKNFVITDPRLPDNPIIFASDSFLQLTEYSREEILGRNCRFLQGPETDRATVRKIRDAIDNQTEVTVQLINYT
KSGKKFWNLFHLQPMRDQKGDVQYFIGVQLDGTEHVRDAAEREGVMLIKKTAEN-IDEAAKRLPDANLAAAAKKKKLD-
GGSGGSGGS-
VSKGEEDNMAIIKEFMRFKVHMEGSVNGHEFEIEGEGEGRPYEGTQTAKLKVTKGGPLPFAWDILSPQFMYGSKAYV
KHPADIPDYLKLSFPEGFKWERVMNFEDGGVVTVTQDSSLQDGEFIYKVKLRGTNFPSDGPVMQKKTMGWEASSER
MYPEDGALKGEIKQRLKLKDGGHYDAEVKTTYKAKKPVQLPGAYNVNIKLDITSHNEDYTIVEQYERAEGRHSTGGM
DELYK-
GAGGT-X-GS-
MPSTRIQQQLGQLTLENLQ

**Extended Data Fig. 7 | Molecular strategy for probing the effect of protein orientation on nuclear export kinetics. (A)** Schematics showing a structural model of the "Export forward" (upper) and "Export reverse" constructs (lower), highlighting the insertion site for the mechanical sensor of interest, from left to right, R16, protein L, Ig27, and Spy0128. **(B)** Amino acid sequence for the "Export forward" and "Export reverse" constructs. Color code indicates each of the molecular components.

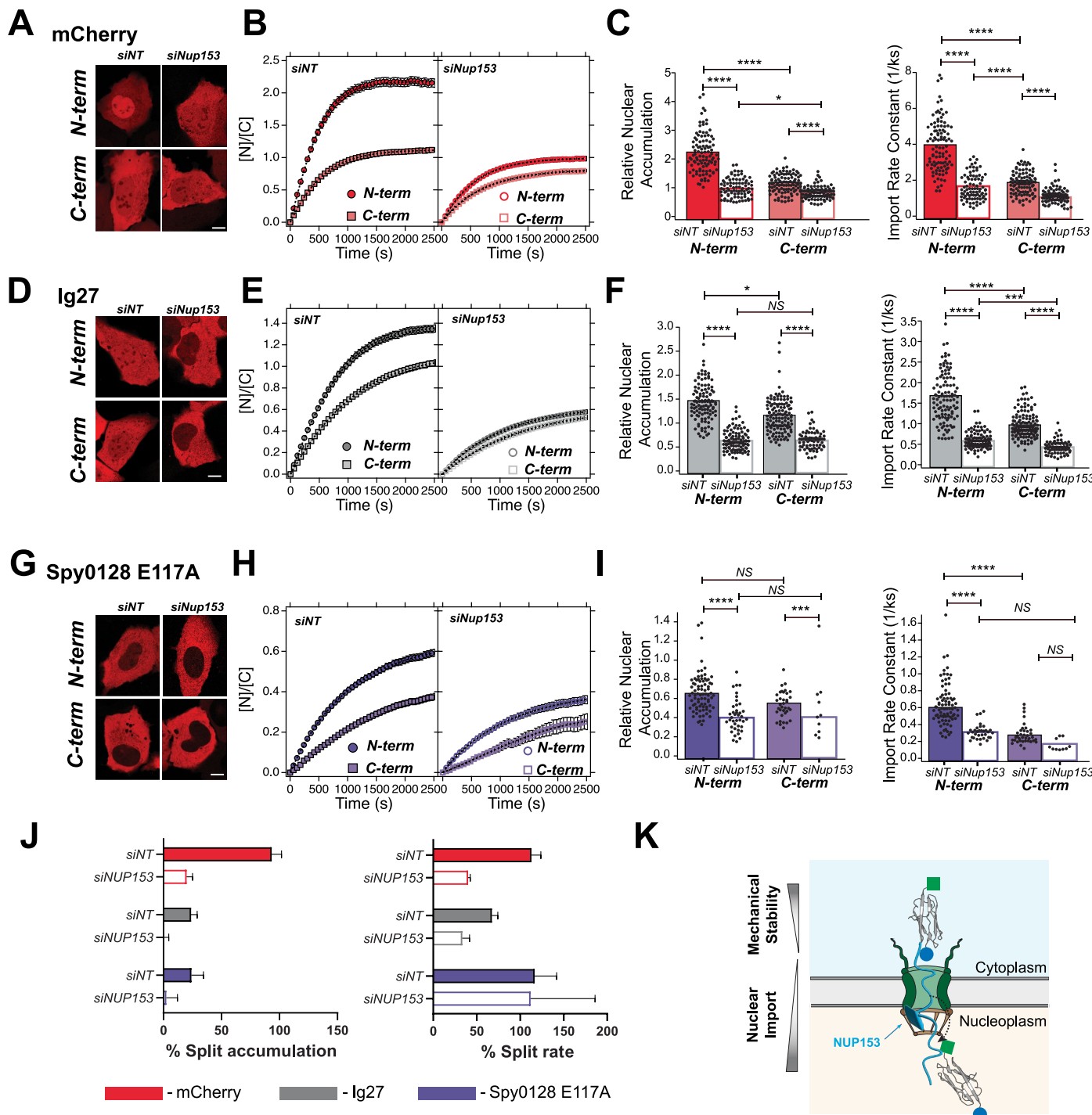

**Extended Data Fig. 8 | See next page for caption.**

**Extended Data Fig. 8 | Downregulation of Nup153 reduces the sensitivity of the NPC to the local mechanical asymmetry of imported protein cargos. (A-C)** Nuclear import kinetics of the mCherry protein when imported from the N- or C-terminus upon downregulating Nup153. (**A**) Representative confocal images of U2OS cells after 20 min of the recovery phase, scale bar 10 μm. (**B**) Average time-courses of the nucleus-to-cytoplasm localisation of the "Import forward" and "Import reverse" mCherry in the control group (*siNT*) and knocking down Nup153 (*siNup153*). (**C**) Relative nuclear accumulation (left) and import rate calculated from fits to the recovery time courses. Bars indicate mean. Accumulation: N-term (*siNT*) $K_e$ = 2.21 ± 0.07; N-term (*siNup153*) $K_e$ = 1.00 ± 0.04; C-term (*siNT*) $K_e$ = 1.14 ± 0.03; C-term (*siNup153*) $K_e$ = 0.83 ± 0.02. Import rate: N-term (*siNT*) $k_1$ = 3.95 ± 0.14 ks$^{-1}$; N-term (*siNup153*) $k_1$ = 1.59 ± 0.08 ks$^{-1}$; C-term (*siNT*) $k_1$ = 0.83 ± 0.02 ks$^{-1}$; C-term (*siNup153*) $k_1$ = 1.13 ± 0.04 ks$^{-1}$. Significance levels for two-tailed Mann-Whitney non-parametric test. Accumulation: N-term (*siNT*) vs. C-term (*siNT*) $P$ = 0; N-term (*siNup153*) vs. C-term (*siNup153*) $P$ = $P$ = 4.00x10$^{-4}$; N-term (*siNT*) vs. N-term (*siNup153*) $P$ = 0; C-term (*siNT*) vs. C-term (*siNup153*) $P$ = 9.9x10$^{-12}$. Import rate: N-term (*siNT*) vs. C-term (*siNT*) $P$ = 0; N-term (*siNup153*) vs. C-term (*siNup153*) $P$ = 3.0x10$^{-6}$; N-term (*siNT*) vs. N-term (*siNup153*) $P$ = 0; N-term (*siNup153*) vs. C-term (*siNup153*) $P$ = 6.7x10$^{-16}$. $n$ = 103 (*siNT* N-term); $n$ = 100 (*siNT* C-term); $n$ = 85 (*siNUP153* N-term); $n$ = 91 (*siNUP153* C-term). (**D-F**) Nuclear import kinetics of the Ig27 when imported from the N- or C- terminus upon downregulating Nup153. (**D**) Representative confocal images of U2OS cells after 20 min of the recovery phase, scale bar 10 μm. (**E**) Average time-courses of the nucleus-to-cytoplasm localisation of the "Import forward" and "Import reverse" Ig27 in the control group (*siNT*) and knocking down Nup153 (*siNup153*). (**F**) Relative nuclear accumulation (left) and import rate calculated from fits to the recovery time courses. Bars indicate mean. Accumulation: N-term (*siNT*) $K_e$ = 1.43 ± 0.04; C-term (*siNT*) $K_e$ = 1.15 ± 0.03; N-term (*siNUP153*) $K_e$ = 0.64 ± 0.02; C-term (*siNup153*) $K_e$ = 0.64 ± 0.02. Import rate: N-term (*siNT*) $k_1$ = 1.63 ± 0.06 ks$^{-1}$; C-term (*siNT*) $k_1$ = 0.97 ± 0.03 ks$^{-1}$; N-term (*siNUP153*) $k_1$ = 0.60 ± 0.02 ks$^{-1}$; C-term

(*siNup153*) $k_1$ = 0.44 ± 0.02 ks$^{-1}$. Significance levels for two-tailed Mann-Whitney non-parametric test. Accumulation: N-term (*siNT*) vs. C-term (*siNT*) $P$ = 5.2x10$^{-9}$; N-term (*siNup153*) vs. C-term (*siNup153*) $P$ = 0.71; N-term (*siNT*) vs. N-term (*siNup153*) $P$ = 0; C-term (*siNT*) vs. C-term (*siNup153*) $P$ = 0. Import rate: N-term (*siNT*) vs. C-term (*siNT*) $P$ = 1.3x10$^{-13}$; N-term (*siNup153*) vs. C-term (*siNup153*) $P$ = 5.2x10$^{-8}$; N-term (*siNT*) vs. N-term (*siNup153*) $P$ = 0; C-term (*siNup153*) vs. C-term (*siNT*) $P$ = 0. $n$ = 103 (*siNT* N-term); $n$ = 101 (*siNT* C-term); $n$ = 85 (*siNUP153* N-term); $n$ = 91 (*siNUP153* C-term). (**G-I**) Nuclear import kinetics of the Spy0128 E117A when imported from the N- or C-terminus upon knocking down Nup153. (**G**) Representative confocal images of U2OS cells after 20 min of the recovery phase, scale bar 10 μm. (**H**) Average time-courses of the nucleus-to-cytoplasm localisation of the "Import forward" and "Import reverse" Spy0128 E117A in the control group (*siNT*) and knocking down Nup153 (*siNup153*). (**I**) Relative nuclear accumulation (left) and import rate calculated from fits to the recovery time courses. Bars indicate mean. Accumulation: N-term (*siNT*) $K_e$ = 0.66 ± 0.22; C-term (*siNT*) $K_e$ = 0.54 ± 0.02; N-term (*siNUP153*) $K_e$ = 0.40 ± 0.03; C-term (*siNup153*) $K_e$ = 0.41 ± 0.06. Import rate: N-term (*siNT*) $k_1$ = 0.63 ± 0.03 ks$^{-1}$; C-term (*siNT*) $k_1$ = 0.29 ± 0.02 ks$^{-1}$; N-term (*siNUP153*) $k_1$ = 0.34 ± 0.02 ks$^{-1}$; C-term (*siNup153*) $k_1$ = 0.16 ± 0.02 ks$^{-1}$. Significance levels for two-tailed Mann-Whitney non-parametric test. Accumulation: N-term (*siNT*) vs. C-term (*siNT*) $P$ = 1.0x10$^{-3}$; N-term (*siNup153*) vs. C-term (*siNup153*) $P$ = 0.40; N-term (*siNT*) vs. N-term (*siNup153*) $P$ = 6.9x10$^{-10}$; C-term (*siNT*) vs. C-term (*siNup153*) $P$ = 4.8x10$^{-3}$. Import rate: N-term (*siNT*) vs. C-term (*siNT*) $P$ = 1.2x10$^{-13}$; N-term (*siNup153*) vs. C-term (*siNup153*) $P$ = 4.9x10$^{-5}$; N-term (*siNT*) vs. N-term (*siNup153*) $P$ = 3.2x10$^{-11}$; C-term (*siNup153*) vs. C-term (*siNT*) $P$ = 1.0x10$^{-3}$. $n$ = 81 (*siNT* N-term); $n$ = 37 (*siNT* C-term); $n$ = *37* (*siNUP153* N-term); $n$ = 10 (*siNUP153* C-term). (**J**) Percentage of directionality in nuclear accumulation (calculated as the ratio between the N-term and C-term accumulation) for the *siNT* control group and *siNup153*. Error bars indicate SEM. (**K**) Schematics of the proposed role of NUP153 in sensing mechanically labile termini to accelerate nuclear transport.

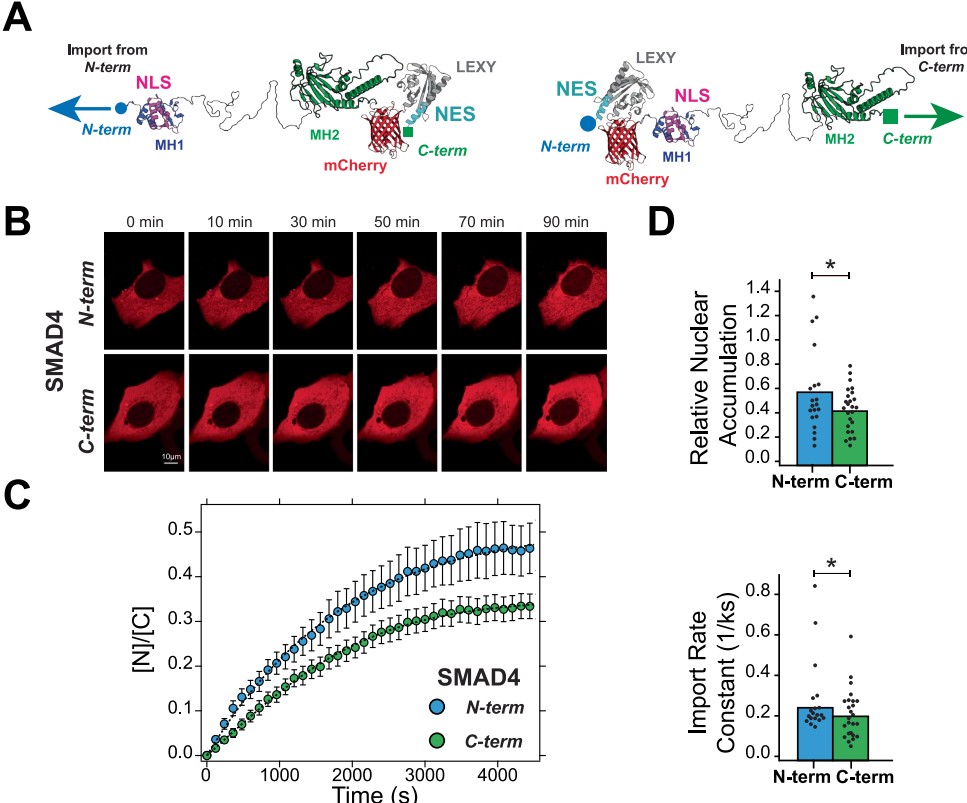

**Extended Data Fig. 9 | SMAD4 shows asymmetric nuclear import when translocating with its N-terminal native NLS. (A)** Schematics of the N-terminal (left) and C-terminal (right) SMAD4 optogenetic constructs, highlighting the position of its native NLS on its N-term MH1 domain. **(B)** Representative confocal images of U2OS cells during the export and recovery phase, scale bar 10 μm. **(C)** Average time-courses of the nucleus-to-cytoplasm localisation of the "Import forward" and "Import reverse" SMAD4 using the native NLS.

**(D)** Relative nuclear accumulation (upper) and import rate (lower) calculated from fits to the recovery time courses. Bars indicate mean. Accumulation: N-term $K_e = 0.56 \pm 0.08$; C-term $K_e = 0.42 \pm 0.04$. Import rate: N-term $k_I = 0.28 \pm 0.04$ ks$^{-1}$; C-term $k_I = 0.21 \pm 0.02$ ks$^{-1}$. Significance levels for two-tailed Mann-Whitney non-parametric test. Accumulation, $P = 4.1 \times 10^{-2}$; Import rate, $P = 3.2 \times 10^{-2}$. $n = 19$ (N-term); $n = 25$ (C-term).

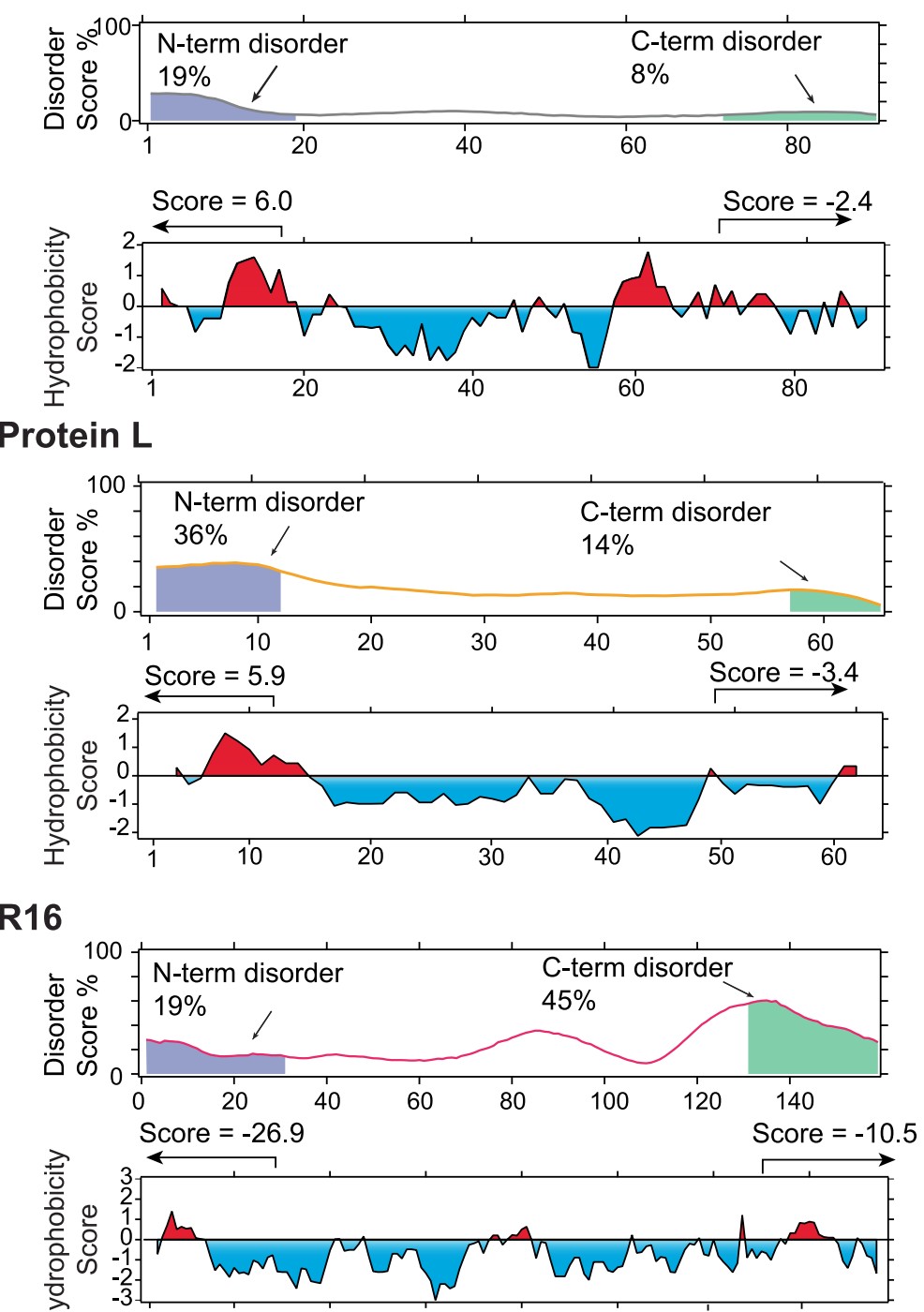

**Extended Data Fig. 10 | SMAD4 shows asymmetric nuclear import when translocating with its N-terminal native NLS. (A)** Schematics of the N-terminal (left) and C-terminal (right) SMAD4 optogenetic constructs, highlighting the position of its native NLS on its N-term MH1 domain. **(B)** Representative confocal images of U2OS cells during the export and recovery phase, scale bar 10 μm. **(C)** Average time-courses of the nucleus-to-cytoplasm localisation of the "Import forward" and "Import reverse" SMAD4 using the native NLS. **(D)** Relative nuclear

accumulation (upper) and import rate (lower) calculated from fits to the recovery time courses. Bars indicate mean. Accumulation: N-term $K_e = 0.56 \pm 0.08$; C-term $K_e = 0.42 \pm 0.04$. Import rate: N-term $k_I = 0.28 \pm 0.04$ ks$^{-1}$; C-term $k_I = 0.21 \pm 0.02$ ks$^{-1}$. Significance levels for two-tailed Mann-Whitney non-parametric test. Accumulation, $P = 4.1 \times 10^{-2}$; Import rate, $P = 3.2 \times 10^{-2}$. $n = 19$ (N-term); $n = 25$ (C-term).

# Reporting Summary

## Statistics

For all statistical analyses, confirm that the following items are present in the figure legend, table legend, main text, or Methods section.

| n/a | Confirmed | |
|---|---|---|
| ☐ | ☒ | The exact sample size (*n*) for each experimental group/condition, given as a discrete number and unit of measurement |
| ☐ | ☒ | A statement on whether measurements were taken from distinct samples or whether the same sample was measured repeatedly |
| ☐ | ☒ | The statistical test(s) used AND whether they are one- or two-sided<br>*Only common tests should be described solely by name; describe more complex techniques in the Methods section.* |
| ☒ | ☐ | A description of all covariates tested |
| ☐ | ☒ | A description of any assumptions or corrections, such as tests of normality and adjustment for multiple comparisons |
| ☐ | ☒ | A full description of the statistical parameters including central tendency (e.g. means) or other basic estimates (e.g. regression coefficient) AND variation (e.g. standard deviation) or associated estimates of uncertainty (e.g. confidence intervals) |
| ☐ | ☒ | For null hypothesis testing, the test statistic (e.g. *F*, *t*, *r*) with confidence intervals, effect sizes, degrees of freedom and *P* value noted<br>*Give P values as exact values whenever suitable.* |
| ☒ | ☐ | For Bayesian analysis, information on the choice of priors and Markov chain Monte Carlo settings |
| ☒ | ☐ | For hierarchical and complex designs, identification of the appropriate level for tests and full reporting of outcomes |
| ☒ | ☐ | Estimates of effect sizes (e.g. Cohen's *d*, Pearson's *r*), indicating how they were calculated |

*Our web collection on statistics for biologists contains articles on many of the points above.*

## Software and code

Policy information about availability of computer code

| Data collection | NIS Elements<br>C++ (Custom written software) |
|---|---|
| Data analysis | Matlab<br>Igor Pro<br>FIJI<br>GraphPad<br>Microsoft Excel |

For manuscripts utilizing custom algorithms or software that are central to the research but not yet described in published literature, software must be made available to editors and reviewers. We strongly encourage code deposition in a community repository (e.g. GitHub). See the Nature Portfolio guidelines for submitting code & software for further information.

## Data

Policy information about availability of data

All manuscripts must include a data availability statement. This statement should provide the following information, where applicable:

- Accession codes, unique identifiers, or web links for publicly available datasets
- A description of any restrictions on data availability
- For clinical datasets or third party data, please ensure that the statement adheres to our policy

The data that support the plots within this paper and other findings of this study are available from the corresponding author upon reasonable request.

## Research involving human participants, their data, or biological material

Policy information about studies with human participants or human data. See also policy information about sex, gender (identity/presentation), and sexual orientation and race, ethnicity and racism.

| | |
|---|---|
| Reporting on sex and gender | N/a |
| Reporting on race, ethnicity, or other socially relevant groupings | N/a |
| Population characteristics | N/a |
| Recruitment | N/a |
| Ethics oversight | N/a |

Note that full information on the approval of the study protocol must also be provided in the manuscript.

# Field-specific reporting

Please select the one below that is the best fit for your research. If you are not sure, read the appropriate sections before making your selection.

☒ Life sciences        ☐ Behavioural & social sciences        ☐ Ecological, evolutionary & environmental sciences

For a reference copy of the document with all sections, see nature.com/documents/nr-reporting-summary-flat.pdf

# Life sciences study design

All studies must disclose on these points even when the disclosure is negative.

| | |
|---|---|
| Sample size | No sample size calculations were performed. All experiments in this work were repeated at least three times. Sample size was chosen based on previous experience and work in the field |
| Data exclusions | Criteria for data exclusions is described in detail in the Materials and Methods section |
| Replication | All experiments were successfully reproduced |
| Randomization | No randomization of experimental groups was performed |
| Blinding | No blinding was used |

# Reporting for specific materials, systems and methods

We require information from authors about some types of materials, experimental systems and methods used in many studies. Here, indicate whether each material, system or method listed is relevant to your study. If you are not sure if a list item applies to your research, read the appropriate section before selecting a response.

## Materials & experimental systems

| n/a | Involved in the study |
|---|---|
| ☐ | ☒ Antibodies |
| ☐ | ☒ Eukaryotic cell lines |
| ☒ | ☐ Palaeontology and archaeology |
| ☒ | ☐ Animals and other organisms |
| ☒ | ☐ Clinical data |
| ☒ | ☐ Dual use research of concern |
| ☒ | ☐ Plants |

## Methods

| n/a | Involved in the study |
|---|---|
| ☒ | ☐ ChIP-seq |
| ☒ | ☐ Flow cytometry |
| ☒ | ☐ MRI-based neuroimaging |

## Antibodies

| Antibodies used | Nup153: Ar301-788A-T (Bethyl Laboratories); GAPDH, Abcam; importin α3 (MAB8204, R&D systems) |
|---|---|
| Validation | All antibodies used were validated by the manufacturer |

## Eukaryotic cell lines

Policy information about cell lines and Sex and Gender in Research

| Cell line source(s) | U2OS, HeLa, and NIH 3T3 |
|---|---|
| Authentication | Cell lines were authenticated by the Cell Services STP in the Francis Crick Institute |
| Mycoplasma contamination | All cell lines were routinely tested for mycoplasma by the Francis Crick institute Cell Services STP |
| Commonly misidentified lines (See ICLAC register) | No cell lines used in this study were commonly misidentified cell lines |

## Plants

| Seed stocks | N/a |
|---|---|
| Novel plant genotypes | N/a |
| Authentication | N/a |

