## [Peer Review File · Nature Physics]

The local mechanostructural properties of protein cargos regulate nucleocytoplasmic transport

Corresponding Author: Professor Sergi Garcia-Manyes

This manuscript has been previously reviewed at another journal. This document only contains information relating to versions considered at Nature Physics.

Version 0:

Reviewer comments:

Reviewer #1

(Remarks to the Author)

The authors did a great job of addressing questions raised in the Reviews. It is a comprehensive, technical tour-de-force. I think their paper will be a great addition to the field.

Reviewer #4

(Remarks to the Author)

I reviewed the resubmission of this manuscript. My comments were properly addressed.

Reviewers comments:

Reviewer #1:

Remarks to the Author:

The authors did a great job of addressing questions raised in the Reviews.

It is a comprehensive, technical tour-de-force.

I think their paper will be a great addition to the field.

R: We thank the reviewer for their thorough review, which has greatly improved our manuscript.

Reviewer #4:

Remarks to the Author:

I reviewed the resubmission of this manuscript. My comments were properly addressed.

R: We thank the reviewer for their review and positive decision.